# Vocal cues to eyewitness accuracy are detected by listeners with and without language comprehension
Philip Urban Gustafsson [1] ✉, Petri Laukka [1,2], Hillary Anger Elfenbein[3] & Nutankumar S. Thingujam [4]

Criminal-law workers must evaluate the accuracy of eyewitness testimony, which includes nonverbal speech characteristics that may carry clues. Using a large collection of testimony statements ($N = 3344$), Study 1 showed that eyewitnesses use a higher speech rate, fewer pauses, and greater acoustic amplitude when remembering correctly and more confident in their answers. In Study 2, observers with vs. without language comprehension (Swedish vs. American and Indian samples) judged the accuracy of testimony statements and judged the eyewitness' confidence. Participants performed at above-chance levels regardless of nation or language comprehension, suggesting universal cues to accuracy. Participants gave higher confidence ratings to correct statements, and this calibration was higher with greater cultural proximity to the testimony source. Observers' confidence judgments were significantly correlated with amplitude, which suggests they used valid acoustical cues. Taken together, results demonstrate distinct vocal markers of eyewitness accuracy, which are detected by listeners from diverse cultures.

There are many known cases whereby eyewitnesses have been wrong in their recall or identification, which has led to innocent convictions[1], making research on witness accuracy important for social justice. Recent work has identified some acoustic cues to accurate witness testimony—notably a higher speech rate[2] — and we attempt to replicate and extend these findings vis-à-vis their potential universality and cultural differences. This is particularly important in multicultural societies with diverse sources of judges, jurors, defendants, and witnesses. We compare accuracy for judges listening to eyewitness testimony in their own vs. a foreign language and among foreign languages for those from more vs. less culturally similar nations. Further, we extend into the eyewitness setting a more general association between vocal characteristics and perceived speaker confidence[3–5], which could be used to evaluate testimony from the eyewitness' perspective.

Many factors can affect witness memory[6], such as exposure duration[7] and leading questions[8]. A general message from this work is that memories will be most accurate when an event has been encoded clearly, with full attention and with a recall following shortly after the event. However, the person judging the accuracy of a witness testimony seldomly knows the exact conditions regarding the event and that which has transpired afterward, and it is further likely that some details will be incorrect even under favorable conditions. This makes it important to investigate cues to recognize when eyewitnesses are likely to be accurate. Retrieval ease and witness confidence are the two cues to

eyewitness accuracy that have received the most attention in previous research. A plethora of studies support the idea that correct memories are generally retrieved more quickly and with less verbal effort than incorrect memories[9–13]. Moreover, the witnesses generally rate themselves as more confident when their memories are correct (albeit not always), in both eyewitness testimony and line-up identifications[10,14–17]. Recently, researchers demonstrated that there are also nonverbal vocal cues to accuracy; correct statements in eyewitness testimony were uttered with *a higher pitch, greater energy in the first formant region, higher speech rate,* and *shorter pauses*[2]. However, more studies are needed to assess the reliability of these initial findings and to examine whether people can detect that these are vocal cues to accuracy.

Although there are few studies on vocal characteristics and eyewitness accuracy[2,18], considerable evidence exists on the relationship between vocal characteristics and confidence more generally. Specifically, studies find that speech is judged (and produced) as more persuasive and confident when uttered with a higher speech rate (including shorter pauses[3,5,19–22]), greater amplitude[4,5,21,22], and a greater amplitude variation[4,20]. Regarding pitch, findings have been inconsistent, with some studies finding that confident speech is uttered with a higher pitch[5,22] and others that confident speech is uttered with a lower pitch[3,4,19]. It is worthwhile to examine vocal characteristics and confidence in an eyewitness testimony setting to determine to what extent these results generalized to a forensic setting.

[1]Department of Psychology, Stockholm University, Stockholm, Sweden. [2]Department of Psychology, Uppsala University, Uppsala, Sweden. [3]Olin Business School, Washington University in St. Louis, St. Louis, MO, USA. [4]Department of Psychology, Tripura University, Agartala, India.
✉e-mail: philipgustafssonresearch@gmail.com

This paper asks two research questions: First, do speakers use different vocal cues depending on their level of confidence? Second, do listeners perceive these potential vocal cues to confidence? The second question is especially important from a forensic perspective because observers' judgments are influenced by the target's nonverbal cues[23] but there are potentially devastating consequences if interpreted incorrectly. For example, a recent study[24] examined witnesses testifying in their native or non-native language and found that non-native speakers were rated by observers as less credible, despite being equally accurate as native speakers. Although the effect sizes were relatively modest, the findings suggest that reliable eyewitness testimony might be regarded inaccurately as low in credibility. Thus, an important unanswered question is how observers rate eyewitness testimony speech across cultures and languages. Given that languages differ in their prosody (i.e., intonation, stress, and timing[25,26], observers' perceived confidence of the speaker could differ depending on their level of comprehension in the spoken language. We expect there could be a greater mismatch between observer-perceived and self-judged confidence the greater the cultural distance between the speaker and listener, in keeping with studies on nonverbal communication of emotions that show listeners are better at recognizing emotions expressions from speakers from their own culture compared to speakers from another culture[27,28].

More specifically, in Study 1, we first attempt to replicate that there are valid vocal acoustic cues to eyewitness accuracy on a larger scale than in previous work. Based on recent research[2], we hypothesize that correct vs. incorrect statements in eyewitness testimony will be uttered with a *higher pitch* (H1), *greater energy in the first formant region* (H2), *higher speech rate* (H3), and *shorter pauses* (H4). Moreover, we examine vocal characteristics related to confidence, and hypothesize that statements judged with high vs. low confidence will be uttered with a *higher speech rate* (H5), *shorter pauses* (H6), *greater amplitude* (H7), *greater amplitude variation* (H8), and *greater pitch variability* (H9). We also explore the relation between confidence and pitch, without specific hypothesis given conflicting results in previous research. In Study 2, we continue by including observers who judge these vocal cues, notably the accuracy and confidence of eyewitness testimony. We hypothesize that observers will judge witness accuracy at above-chance levels (H10a), and that native speakers (Swedish) will be more accurate than culturally similar non-native speakers (American), who in turn will be more accurate than more culturally distant non-native speakers (Indian; H10b). We also hypothesize that observers will give higher confidence ratings to correct versus incorrect statements (H11a), again with greater effects for Swedish speakers, followed by the non-Swedish groups (H11b). We further expect that observer-judged confidence will correlate with witnesses' own self-reported confidence (H12a), with a greater association for Swedish participants, followed by American and finally Indian participants (H12b). On an exploratory basis, we also examine relations of observer-rated confidence with a comprehensive set of acoustic cues, to observe the vocal cues that participants use when judging confidence.

## Methods
### Transparency and openness
Hypotheses were preregistered for both Study 1 (January 18th, 2023; https://osf.io/ahqze) and Study 2 (March 16th, 2023; https://osf.io/9zvnm). Data and code for Study 1 are publicly accessible at https://osf.io/x7u5g and for Study 2 at https://osf.io/x7u5g. We report how we determined our sample size, all data exclusions (if any), all manipulations, and all measures in the study, and the study follows JARS[29]. Data were analyzed using Rstudio[30] in R version 4.0.5[31]. Multilevel analyses were carried out with the *lme4*-package[32], and figures were created using the *tidyverse*-package[33]. All statistical tests were two-sided with alpha level set to 0.05 for statistical significance. Study 1 was approved by the Swedish Ethical Review Authority (#2018/2030-31/5). Study 2 was approved by the Swedish Ethical Review Authority (#2018/2030-31/5) for data collected in Sweden and by the Institutional Review Board at Washington University in St. Louis for data collected in the USA. Ethical vetting was not performed for the Indian data collection as it was not required according to the guidelines for psychological research at Tripura University. Data distribution was assumed to be normal, but this was not formally tested.

## Study 1
### Participants and materials.
Stimuli were drawn from a previous study[10], in which 56 participants ($M_{age}$ = 29.45, $SD_{age}$ = 8.22; 38 women, 18 men) with normal or corrected-to-normal vision watched a staged crime film and were interviewed as eyewitnesses. All participants provided informed consent to participate. The current dataset contains data from 51 participants ($M_{age}$ = 29.45, $SD_{age}$ = 8.23; 34 women, 17 men) because three participants did not consent to further use of their data, and data from two participants contained excessive background noise that prevented meaningful audio from being extracted (see more about statement exclusion below). The resulting dataset contained 3344 statements (76.61% accurate; $M_{duration}$ = 4.26 seconds, $SD$ = 3.13 seconds). No a priori power analysis was calculated, as we were constrained by available data. However, a sensitivity power analysis suggested 80% power to detect at least $d$ = .07, given a standard 0.05 alpha error probability and a two-tailed test.

Participants were interviewed twice, with the first interview taking place straight after viewing the staged crime film and the second interview taking place two weeks later. Only data from the first interview session is analyzed in this study. The interview started with a free-recall phase ("I would like you to start by freely recounting what you have seen") and was followed by cued-recall questions (e.g., "How old was the perpetrator?"). During the course of the interview, the interviewer wrote down details mentioned by the participant on an answer sheet. After the interview, the interviewer read these details aloud, and the participant rated confidence in each detail (0–100% with integers of 20). The interviews lasted between 7 min and 43 seconds to 17 min and 36 seconds (median length = 11 min and 23 seconds), and were videotaped and transcribed verbatim. Coders split each interview into separate, verifiable statements (e.g., "I think he was around 30 years old"; exact overlap = 82.19 %) that were then coded for accuracy (correct/incorrect; exact overlap = 82.15%, $\kappa$ = 0.69).

The dataset was prepared for analysis first by converting each testimony from video to audio format, using the converter function of the VLC Media Player (v. 3.0.12). Each witness testimony was then cut into single audio clips for each testimony statement, using Audacity (v.2.4.2). They were cut so that they either started as soon as the interviewer had finished their question, or as soon as the witness had finished talking about the previous statement, and ended as soon as the last syllable was produced in the current statement. This process was made possible by comparing sound wave intensity, visible in Audacity. Some testimony statements were uttered concurrently with interviewer speech and or concurrently with other noise (partial roadwork occurred outside during some of the interviews), and therefore, had to be excluded.

### Recording and acoustic analysis.
Videos were recorded with a Canon HF100 [E] camcorder set on a tripod in a laboratory with brick walls and a small window.

The audio files were analyzed along 16 acoustic dimensions. These dimensions were determined in a previous study[2] by running a principal-component analysis on 88 acoustic dimensions, as well as by extrapolating findings from confidence speech research. These acoustic dimensions represent different aspects of *frequency* (e.g., fundamental frequency, which is related to perceived pitch), *energy* (related to perceived loudness), *spectral balance* (related to perceived voice quality), and *temporal characteristics* (e.g., speech rate and pauses) of the voice signal. Table 1 shows a description of each acoustic dimension (i.e., vocal characteristic). To obtain values for each of these dimensions, we ran each audio clip through the OpenSMILE software[34] using the extended GeMAPS parameter set[35]. Missing values ($n$ = 0.3%) were replaced by imputing the weighted mean for correct and incorrect statements within each participant and vocal characteristic.

## Study 2
### Participants.
Two-hundred and seventy-seven participants took part in Study 2 ($M_{age}$ = 24.27, SD = 11.38; 153 women, 120 men, 1 non-binary, 3 non-responses). Participants were recruited from three different countries, Sweden ($n$ = 61, $M_{age}$ = 37.02, SD = 11.78; 40 women, 17 men, 1 non-binary,

**Table 1 | Summary of Selected Vocal Characteristics**

| Feature type and abbreviation | Description |
|---|---|
| *Temporal cues* | |
| VoicedSegPerSec | The number of continuous voiced regions per second (pseudo syllable rate) |
| VoicedSegM | Mean length of continuously voiced regions |
| UnvoicedSegM | Mean length of unvoiced regions (approximating pause duration) |
| *Spectral balance cues* | |
| F1 (amp) | Relative energy of the spectral envelope in the first formant region |
| AlphaRatio_UV | Ratio of the summed energy from 50-1000 Hz and 1000-5000 Hz, for unvoiced regions |
| H1-H2 | Ratio of energy of the first F0 harmonic (H1) to the energy of the second F0 harmonic (H2) |
| MFCC3_V (M) | Mean of the third Mel-Frequency Cepstral Coefficient, for voiced regions |
| SpectralSlope_V | Mean spectral slope (i.e. linear regression slope of the logarithmic power spectrum) for the 500–1500 Hz region, for voiced regions |
| *Energy cues* | |
| Int (M) | Mean voice intensity estimated from an auditory spectrum |
| Int (SD) | Standard deviation (normalized by arithmetic mean) of voice intensity estimated from an auditory spectrum |
| *Frequency cues* | |
| F0 (M) | Mean fundamental frequency (F0) on a semitone frequency scale, starting at 27.5 Hz (semitone 0) |
| F0 (SD) | Standard deviation (normalized by the arithmetic mean) of fundamental frequency |
| F0 (rise) | Mean slope of signal parts with rising F0 |
| F0 (fall) | Mean slope of signal parts with falling F0 |
| F1 (M) | Mean of first formant (F1) center frequency |
| F3 (bw) | Mean bandwidth of the third formant (F3) |

A detailed description of the vocal characteristics is available in previous research[2,34].

3 non-responses), USA ($n = 156$, $M_{age} = 19.40$, SD = 3.20; 83 women, 73 men) and India ($n = 60$, $M_{age} = 19.40$, SD = 0.91; 30 women, 30 men).

All participants provided informed consent to participate. Participants from Sweden were recruited from an online recruitment pool and received a movie voucher as compensation; participants from USA were recruited at the Washington University in St. Louis and took part in the study for course credit, and participants from India were recruited from Tripura University and received a monetary compensation of 500 Indian rupees (around 5.6 euro). In the Indian sample, the participants were Bengali (27), Assamese (9), Indian Nepali (1), and tribes of Tripura (18), Manipuri (3), and from other north Indian groups (2). In the American sample, there were 53 Asian or Pacific Islanders, 13 Black or African Americans, 9 Hispanic or Latinos, 75 White or Caucasians, 2 Biracial or Multiracial, and 4 who rated "a race/ethnicity not listed". All participants in the American and Indian samples reported that they had no Swedish language comprehension skills.

An a priori power analysis for one-way fixed effects analysis of variance (ANOVA) with 3 groups suggested $n = 159$, given 80% power to detect a medium effect size (f = 0.25) at standard 0.05 alpha error probability. We chose a medium effect size in order to focus on the most robust effects while still having enough power also to detect effects that could be smaller yet still have practical and theoretical significance. Please note that this calculation deviates from the preregistration, wherein the medium effect size was erroneously entered as f = 0.39. We note that our final sample was larger than planned because a greater number of participants than anticipated signed up in the USA. Due to guidelines regarding course credit, all interested participants were welcomed into the study, and we did not exclude otherwise eligible data from analysis.

**Materials and procedure.** Stimuli consisted of a selection of the audio clips from Study 1. Specifically, we randomly sampled 5 correct and 5 incorrect responses to cued recall questions from each of the witness testimony statements used in Study 1. Out of the 51 testimonies, one contained fewer than 5 incorrect statements ($n = 4$), and was therefore excluded from the sample, leaving us with a pool of 500 witness statements (50% correct). To prevent fatigue for participants, we split these samples into five sets, grouping

the 500 statements into five groups of 100 statements that each contained all sampled statements from 10 witnesses in each group. Participants were then randomly assigned into one of five surveys containing one of these pools of 100 statements. The presentation order of the statements was randomized. All statements were spoken in Swedish and lasted 0.36–21.45 seconds ($M = 4.44$, SD = 3.34).

The survey was created using Qualtrics, and participants completed it online by accessing a link provided by the researcher. Participants were first informed that the purpose of the study was to examine judgments of eyewitness accuracy. They were also informed that the witnesses had watched a staged crime film and gave sincere testimony about what they had seen, and that the task was not about detecting deception, but instead to detect when a statement was correct or incorrect. Furthermore, all participants were informed that they should focus on the "nonverbal aspects of the speech, such as the tone of voice" rather than on the verbal content when making their judgments. The American and Indian participants were additionally informed that the statements were spoken in Swedish and they were not expected to understand the content of the speech. Next, participants went through an audio check to make sure they could hear the audio files. They were then instructed that they would listen to statement excerpts from testimony one-by-one, and that their task was to 1) judge accuracy (correct/incorrect) and 2) judge perceived speaker confidence (Likert scale ranging from 1 "Not confident at all" to 9 "Very confident"). After judging their pool of 100 statements, participants provided demographic information and were thanked for participation.

**Reporting summary**
Further information on research design is available in the Nature Portfolio Reporting Summary linked to this article.

## Results
### Study 1 – accuracy
We started by examining to what extent correct and incorrect statements differed along the vocal characteristics. This was done while combining free and cued recall, given that we did not expect any interaction effects between retrieval mode and accuracy (separate plotting of the data for free and cued

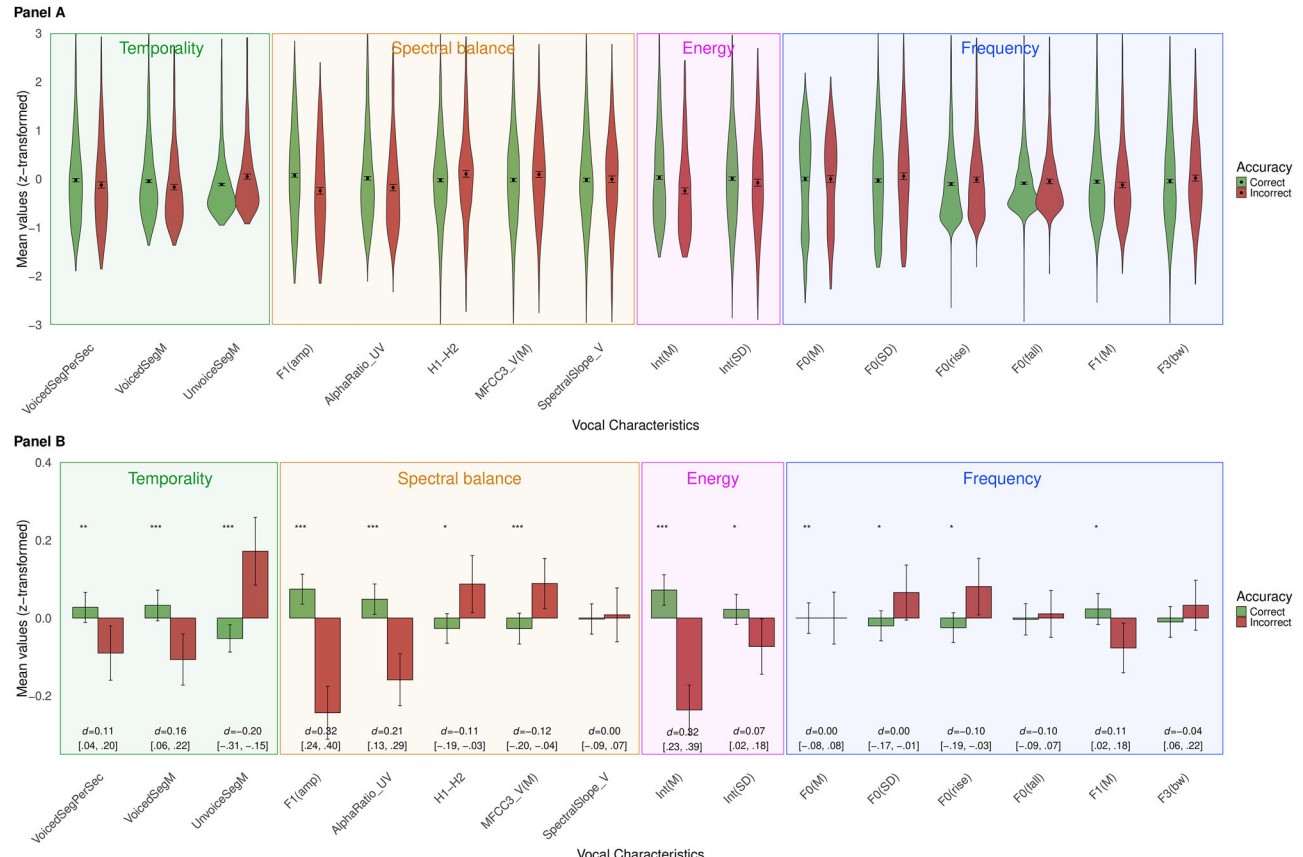

**Fig. 1 | Vocal Characteristics for Correct and Incorrect Statements.** Legend: Panel **A** displays violin plots showing the distribution of data, with mean and 95% confidence intervals. Panel **B** zooms in on the means and 95% CIs, without showing the distribution, for a clearer display of the patterns of acoustic parameters across conditions. $n = 3344$. $d$ = Cohen's d, *$p < .05$, **$p < .01$, ***$p < 001$. Exact $p$-values are available in Table 2.

recall are available as supplementary material [Supplementary Fig. 1] and indeed show largely identical patterns). For this analysis, multilevel modelling (lme4-package[32]), with statements nested within participants (i.e., random factor) was used. We first created baseline, intercept-only models with each vocal characteristic as the outcome variable, and compared these with models containing accuracy as a predictor.

Results supported H2 – *greater energy in the first formant region* (F1 [amp]), H3 - *higher speech rate* (VoicedSegPerSec); and H4 – *shorter pauses* (UnvoicedSeg[M]; see Fig. 1 and Table 2). Although the multilevel analysis showed a statistically significant effect of accuracy on *mean pitch* (F0 [M]), H1 predicted a higher pitch for correct statements, and an inspection of the means in Fig. 1 suggests no meaningful difference in pitch between correct and incorrect statements. The effect size (Cohen's d) is effectively zero.

In addition to these hypothesized results, the multilevel analyses showed that correct statements were uttered with *lower pitch variation* (F0 [SD]), *lower mean rising slope* (F0 [rise]), *higher pitch in the first formant region* (F1[M]), *greater amplitude* (Int [M]), *greater amplitude variation* (Int [SD]), *higher alpha ratio for unvoiced regions* (AlphaRatio_UV), *lower energy in the H1-H2 range* (H1-H2), *lower mean of the third Mel-Frequency Cepstral Coefficient, for voiced regions* (MFCC3_V [M]), and *longer duration of voiced regions* (VoicedSeg [M]); see Fig. 1 and Table 2. Descriptive statistics for all vocal characteristics, for both correct and incorrect statements, are shown in Supplementary Table 1.

**Study 1 – confidence**
Next, we examined the association between confidence and vocal characteristics. This analysis used the subset of the data ($n = 2428$) including all statements for which confidence judgments were available. Based on analysis of Pearson correlations, there were several statistically significant effects, but the effect sizes were small ($r_{max} = 0.25$; see Table 3). Given that these small effects could result from a restriction of range in the data (75.28% of all statements were judged with a confidence of 80–100%), we conducted an additional analysis with dummy-codes for confidence into "maximum confidence" (100% confidence) and "less than maximum confidence" (<100% confidence). Using this variable, we ran multilevel analyses that were similar to how models were run for accuracy and vocal characteristics, with statements nested within participants.

Results supported H5, as statements judged by the witness themselves with "maximum confidence" were uttered with a *higher speech rate* (VoicedSegPerSec); H6 - *shorter pauses* (UnvoicedSeg [M]); and H7 - *greater amplitude* (Int [M]; see Fig. 2 & Table 2). We found no support for H8, as there was no statistically significant difference for amplitude variation (Int [SD]). Likewise, for H9, pitch variability (F0 [SD]) showed a smaller rather than greater variability for statements rated by the witness with "maximum confidence".

In addition to our hypotheses, in exploratory results we found that statements with "maximum confidence" were uttered with a *lower pitch* (F0 [M]), *higher pitch in the first formant region* (F1 [M]), *greater energy in the first formant region* (F1[amp]), *greater energy in higher vs. lower frequencies of the spectrum for unvoiced sounds* (AlphaRatio_UV), *lower mean of the third Mel-Frequency Cepstral Coefficient, for voiced regions* (MFCC3_V[M]), *lower spectral slope in the 500–1500 Hz region, for voiced regions* (SpectralSlope_V), and *longer duration of voiced regions* (VoicedSegM); see Fig. 2 and Table 2.

**Study 2 – judgment accuracy**
We first examined the accuracy of participants' judgments to determine the extent to which it exceeded the chance level. We measured accuracy in the form of discrimination accuracy ($d'$) from the signal detection framework[36]. This is a combined score of the ability to detect a *correct* statement and the

**Table 2 | Model Comparisons of the Effects of Memory Accuracy and Confidence on Vocal Characteristics**

| | Accuracy | Confidence |
|---|---|---|
| *Temporal cues* | | |
| VoicedSegPerSec | $\chi^2(1, N = 3344) = 8.03$, $p = 0.005$, $wi(AIC) = 0.95$ | $\chi^2(1, N = 2428) = 37.40$, $p < 0.001$, $wi(AIC) > 0.99$ |
| VoicedSegM | $\chi^2(1, N = 3344) = 19.82$, $p < 0.001$, $wi(AIC) > 0.99$ | $\chi^2(1, N = 2428) = 43.27$, $p < 0.001$, $wi(AIC) > 0.99$ |
| UnvoicedSegM | $\chi^2(1, N = 3344) = 31.86$, $p < 0.001$, $wi(AIC) > 0.99$ | $\chi^2(1, N = 2428) = 108.13$, $p < 0.001$, $wi(AIC) > 0.99$ |
| *Spectral balance cues* | | |
| F1 (amp) | $\chi^2(1, N = 3344) = 100.88$, $p < 0.001$, $wi(AIC) > 0.99$ | $\chi^2(1, N = 2428) = 206.57$, $p < 0.001$, $wi(AIC) > 0.99$ |
| AlphaRatio_UV | $\chi^2(1, N = 3344) = 30.24$, $p < 0.001$, $wi(AIC) > 0.99$ | $\chi^2(1, N = 2428) = 64.19$, $p < 0.001$, $wi(AIC) > 0.99$ |
| H1-H2 | $\chi^2(1, N = 3344) = 7.25$, $p = 0.007$, $wi(AIC) = 0.95$ | $\chi^2(1, N = 2428) = 3.05$, $p = .081$, $wi(AIC) = 0.27$ |
| MFCC3_V (M) | $\chi^2(1, N = 3344) = 12.49$, $p < 0.001$, $wi(AIC) > 0.99$ | $\chi^2(1, N = 2428) = 35.26$, $p < 0.001$, $wi(AIC) > 0.99$ |
| SpectralSlope_V | $\chi^2(1, N = 3344) = 0.23$, $p = 0.635$, $wi(AIC) = 0.27$ | $\chi^2(1, N = 2428) = 6.27$, $p = 0.012$, $wi(AIC) = 0.88$ |
| *Energy cues* | | |
| Int (M) | $\chi^2(1, N = 3344) = 93.34$, $p < 0.001$, $wi(AIC) > 0.99$ | $\chi^2(1, N = 2428) = 203.14$, $p < 0.001$, $wi(AIC) > 0.99$ |
| Int (SD) | $\chi^2(1, N = 3344) = 6.78$, $p = 0.009$, $wi(AIC) = 0.92$ | $\chi^2(1, N = 2428) = 0.37$, $p = 0.545$, $wi(AIC) = 0.30$ |
| *Frequency cues* | | |
| F0 (M) | $\chi^2(1, N = 3344) = 7.16$, $p = 0.007$, $wi(AIC) = 0.92$ | $\chi^2(1, N = 2428) = 0.08$, $p = 0.776$, $wi(AIC) = 0.27$ |
| F0 (SD) | $\chi^2(1, N = 3344) = 6.15$, $p = 0.013$, $wi(AIC) = 0.89$ | $\chi^2(1, N = 2428) = 5.06$, $p = 0.025$, $wi(AIC) = 0.82$ |
| F0 (rise) | $\chi^2(1, N = 3344) = 5.50$, $p = 0.019$, $wi(AIC) = 0.88$ | $\chi^2(1, N = 2428) = 1.86$, $p = 0.172$, $wi(AIC) = 0.50$ |
| F0 (fall) | $\chi^2(1, N = 3344) = 0.11$, $p = 0.744$, $wi(AIC) = 0.27$ | $\chi^2(1, N = 2428) = 3.30$, $p = 0.069$, $wi(AIC) = 0.62$ |
| F1 (M) | $\chi^2(1, N = 3344) = 4.99$, $p = 0.025$, $wi(AIC) = 0.73$ | $\chi^2(1, N = 2428) = 7.50$, $p = 0.006$, $wi(AIC) = 0.92$ |
| F3 (bw) | $\chi^2(1, N = 3344) = 0.71$, $p = 0.400$, $wi(AIC) = 0.38$ | $\chi^2(1, N = 2428) = 1.20$, $p = 0.272$, $wi(AIC) = 0.38$ |

See Table 1 for full description of abbreviated vocal characteristics. wi(AIC) = Akaike weights (which represent the relative likelihood of a model). n = 277.

ability to detect an *incorrect* statement. We tested this with one-sample $t$-tests against 0 ($d' = 0$ indicates chance performance). In line with predictions (H10a), all groups performed above chance level; Swedish sample: $t(60) = 3.98$, $p < 0.001$, Cohen's $d = 0.51$, 95% CI [0.24, 0.78]; American sample: $t(155) = 5.94$, $p < 0.001$, Cohen's $d = 0.48$, 95% CI [0.31, 0.64]; Indian sample: $t(59) = 5.09$, $p < 0.001$, Cohen's $d = 0.66$, 95% CI [0.37, 0.94]; with an overall score across groups above chance level: $t(276) = 8.47$, $p < 0.001$, Cohen's $d = 0.51$, 95% CI [0.38, 0.63] (see Table 4). These results show that participants identified the accuracy of the witness' spoken statements better than chance. Table 4 contains discrimination accuracy ($d'$), hit rate, false-alarm rate, and response bias ($c$). There were no statistical tests for response bias, and Table 4 shows that all groups displayed a liberal bias—that is, judges were more likely in general to judge a statement as correct rather than incorrect.

Next, we compared the relative performance across the groups, where we expected the Swedish sample to perform best, followed by the American sample, and then the Indian sample (H10b). Contrary to this prediction results showed no statistically significant effects between the groups on

judgment accuracy: Sweden - USA: $\Delta M = 0.11$, $t(80) = 1.50$, $p = 0.137$, Cohen's $d = 0.29$, 95% CI [−0.01, 0.59]; Sweden - India: $\Delta M = 0.05$, $t(102) = −0.60$, $p = 0.547$, Cohen's $d = 0.12$, 95% CI [−0.24, 0.48]; India - USA: $\Delta M = 0.06$, $t(107) = 1.19$, $p = 0.236$, Cohen's $d = 0.13$, 95% CI [−0.43, 0.17]] (see Table 4). For these non-significant findings, we also calculated Bayes factors (BF): Sweden – USA, BF = 0.75; Sweden – India, BF = 0.23; India – USA, BF = 0.32.

## Study 2 – confidence judgments

After examining judgment accuracy, we examined participants' judgments of confidence. We hypothesized that participants would perceive correct statements as being uttered with higher confidence compared to incorrect statements (H11a), with greater effects for the Swedish sample, followed by the American sample, followed by the Indian sample (H11b). We tested this using an ANOVA with witness accuracy, country and their interaction as predictors, and confidence judgment as outcome. The judgment of confidence was the outcome because it could not have a causal effect on witness accuracy. Supporting H11a, the ANOVA showed a statistically significant effect of Accuracy, $F(1, 22,240) = 220.55$, $p < 0.001$, partial $\eta2 = 0.010$, 90% CI [0.008, 0.012], in which judgments of confidence were higher for correct statements ($M = 5.44$, $SD = 2.23$) than for incorrect statements ($M = 5.00$, $SD = 2.24$). Moreover, there was a statistically significant effect of Country, $F(2, 22,239) = 132.10$, $p < 0.001$, partial $\eta2 = 0.012$, 90% CI [0.009, 0.014], wherein the highest judged confidence came from the Indian sample ($M = 5.56$, $SD = 2.28$), followed by the Swedish sample ($M = 5.42$, $SD = 2.28$), followed by the American sample ($M = 5.01$, $SD = 2.20$). There was also a statistically significant interaction between Accuracy and Country, $F(2, 22,237) = 8.60$, $p < 0.001$, partial $\eta2 = 0.001$, 90% CI [0.0002, 0.0015]. Planned comparisons showed that all participants judged that correct statements were uttered with higher confidence compared to incorrect statements (Sweden: $\Delta M = 0.66$, $t(4810) = 10.15$, $p < 0.001$, Cohen's $d = 0.29$, 95% CI [0.24, 0.35]; USA: $\Delta M = 0.42$, $t(12595) = 10.79$, $p < 0.001$, Cohen's $d = 0.19$, 95% CI [0.16, 0.23]; India: $\Delta M = 0.30$, $t(4831) = 4.58$, $p < 0.001$, Cohen's $d = 0.13$, 95% CI [0.08, 0.19]). Supporting H11b, the difference in judged confidence between correct and incorrect statements was greater for the Swedish sample both compared to the American sample ($\Delta M = 0.26$, $t(215) = 2.75$, $p = 0.008$, Cohen's $d = 0.49$, 95% CI [0.19, 0.79]) and compared to the Indian sample ($\Delta M = 0.36$, $t(119) = 3.23$, $p = 0.002$, Cohen's $d = 0.59$, 95% CI [0.22, 0.95]). There was no statistically significant difference between the American and Indian sample ($\Delta M = 0.10$, $t(214) = 1.27$, $p = 0.183$, Cohen's $d = 0.21$, 95% CI [−0.09, 0.51]). Next, we examined the correlation between participant-judged confidence and witnesses' own confidence. We expected these variables to correlate (H12a), with greater effects for the Swedish sample, followed by the American sample, followed by the Indian sample (H12b). Using Pearson correlations, we found statistically significant correlations for the Swedish sample, $r = 0.189$, 95% CI [0.162, 0.216], $p < 0.001$; and the American sample $r = 0.042$, 95% CI [0.025, 0.059], $p < 0.001$; but not for the Indian sample, $r = 0.014$, 95% CI [−0.014, 0.042], $p = 0.399$ (see Table 3), partially supporting H12a. Overall, there was a statistically significant correlation across groups, $r = 0.069$, 95% CI [0.055, 0.082], $p < 0.001$. Partially supporting H12b, the difference between the Swedish and American samples was significant (Z = 8.81, 95% CI [6.85, 10.77], $p < 0.001$), but the difference between the American and the Indian sample was not statistically significant (Z = 1.66, 95% CI [−0.30, 3.62], $p = 0.098$).

Finally, we explored the correlation between observer's judged confidence of the eyewitness statements and the vocal cues. The results shown in Table 3 indicate similar performance for all groups across vocal cues ($\Delta r_{max} = 0.08$, range = −0.23 to 0.30), with the strongest correlations between observer-judged confidence and *mean amplitude* (Int [M]; $r_{all} = .30$), *amplitude in the first-formant region* (F1 [amp]; $r_{all} = 0.25$), *alpha-ratio for unvoiced regions* (AlphaRatio_UV; $r_{all} = 0.25$) and *shorter pauses* (UnvoicedSeg[M]; $r_{all} = −0.22$).

**Table 3 | Correlation Coefficients (Pearson r) for Vocal Characteristics and Confidence**

| Vocal characteristics | Witness confidence (Study 1) | Observer-judged confidence (Study 2) | | | |
|---|---|---|---|---|---|
| | | Sample | | | |
| | | Sweden | USA | India | All |
| *Temporal cues* | | | | | |
| VoicedSegPerSec | r = 0.11, 95% CI [0.07, 0.15], p < 0.001 | r = 0.15, 95% CI [0.12, 0.18], p < 0.001 | r = 0.20, 95% CI [0.18, 22], p < 0.001 | r = 0.17, 95% CI [0.15, 0.20], p < 0.001 | r = 0.18, 95% CI [0.15, 20], p < 0.001 |
| VoicedSegM | r = 0.13, 95% CI [0.09, 0.17], p < 0.001 | r = 0.07, 95% CI [0.04, 0.09], p < 0.001 | r = 0.08, 95% CI [0.07, 0.10], p < 0.001 | r = 0.06, 95% CI [0.03, 0.09], p < 0.001 | r = 0.08, 95% CI [0.06, 0.09], p < 0.001 |
| UnvoicedSegM | r = −0.20, 95% CI [−0.17, −0.24], p < 0.001 | r = −0.21, 95% CI [−0.18, −0.24], p = 0.001 | r = −0.23, 95% CI [−0.21, −0.25], p < 0.001 | r = −0.20, 95% CI [−0.17, −0.22], p < 0.001 | r = −0.22, 95% CI [−0.20, −0.23], p < 0.001 |
| *Spectral balance cues* | | | | | |
| F1 (amp) | r = 0.25, 95% CI [0.21, 0.29], p < 0.001 | r = 0.22, 95% CI [0.20, 0.25], p < 0.001 | r = 0.26, 95% CI [0.25, 0.28], p < 0.001 | r = 0.23, 95% CI [0.21, 0.26], p < 0.001 | r = 0.25, 95% CI [0.23, 0.26], p < 0.001 |
| AlphaRatio_UV | r = 0.14, 95% CI [0.10, 0.18], p < 0.001 | r = 0.21, 95% CI [0.18, 0.24], p < 0.001 | r = 0.29, 95% CI [0.27, 0.30], p < 0.001 | r = 0.24, 95% CI [0.22, 27], p < 0.001 | r = 0.26, 95% CI [0.25, 0.27], p < 0.001 |
| H1-H2 | r = −0.01, 95% CI [−0.05, 0.03], p = 0.628 | r = −0.05, 95% CI [−0.02, −0.07], p = 0.001 | r = −0.10, 95% CI [−0.08, −0.12], p < 0.001 | r = −0.08, 95% CI [−0.05, −0.11], p < 0.001 | r = −0.08, 95% CI [−0.07, −0.10], p < 0.001 |
| MFCC3_V (M) | r = −0.07, 95% CI [−0.03, −0.11], p < 0.001 | r = −0.06, 95% CI [−0.03, −0.09], p = 0.000 | r = −0.10, 95% CI [−0.08, −0.12], p < 0.001 | r = −0.10, 95% CI [−0.07, −0.13], p < 0.001 | r = −0.09, 95% CI [−0.08, −0.10], p < 0.001 |
| SpectralSlope_V | r = −0.05, 95% CI [−0.01, −0.09], p = 0.012 | r = −0.13, 95% CI [−0.10, −0.16], p < 0.001 | r = −0.15, 95% CI [−0.13, −0.17], p < 0.001 | r = −0.14, 95% CI [−0.11, −0.17], p < 0.001 | r = −0.14, 95% CI [−0.13, −0.16], p < 0.001 |
| *Energy cues* | | | | | |
| Int (M) | r = 0.20, 95% CI [0.16, 0.24], p < 0.001 | r = 0.25, 95% CI [0.23, 0.28], p < 0.001 | r = 0.32, 95% CI [0.31, 0.34], p < 0.001 | r = 0.30, 95% CI [0.27, 0.32], p < 0.001 | r = 0.30, 95% CI [0.29, 0.31], p < 0.001 |
| Int (SD) | r = −0.02, 95% CI [−0.06, 0.02], p = 0.398 | r = 0.09, 95% CI [0.06, 0.12], p < 0.001 | r = 0.11, 95% CI [0.09, 0.13], p < 0.001 | r = 0.13, 95% CI [0.10, 0.15], p < 0.001 | r = 0.11, 95% CI [0.09, 0.12], p < 0.001 |
| *Frequency cues* | | | | | |
| F0 (M) | r = −0.06, 95% CI [−0.02, −0.10], p = 0.002 | r = 0.02, 95% CI [−0.01, 0.05], p = 0.133 | r = 0.02, 95% CI [0.00, 0.03], p = 0.078 | r = 0.01, 95% CI [−0.01, 0.04], p = 0.350 | r = 0.02, 95% CI [0.00, 0.03], p = 0.013 |
| F0 (SD) | r = −0.10, 95% CI [−0.06, −0.14], p < 0.001 | r = 0.03, 95% CI [0.00, 0.06], p = 0.045 | r = 0.05, 95% CI [0.03, 0.07], p < 0.001 | r = 0.04, 95% CI [0.01, 0.07], p = 0.006 | r = 0.04, 95% CI [0.03, 0.06], p < 0.001 |
| F0 (rise) | r = −0.06, 95% CI [−0.02, −0.10], p = 0.006 | r = 0.00, 95% CI [−0.03, 0.03], p = 0.873 | r = 0.00, 95% CI [−0.02, 0.02], p = 0.861 | r = 0.01, 95% CI [−0.02, 0.03], p = 0.682 | r = 0.00, 95% CI [−0.01, 0.01], p = 0.977 |
| F0 (fall) | r = −0.07, 95% CI [−0.03, −0.11], p < 0.001 | r = 0.03, 95% CI [0.00, 0.06], p = 0.056 | r = 0.01, 95% CI [−0.01, .03], p = 0.186 | r = 0.02, 95% CI [0.00, 0.05], p = 0.090 | r = 0.02, 95% CI [0.00, 0.03], p = 0.008 |
| F1 (M) | r = 0.08, 95% CI [0.04, 0.12], p < 0.001 | r = 0.09, 95% CI [0.06, 0.12], p < 0.001 | r = 0.03, 95% CI [0.01, 0.04], p = 0.004 | r = 0.07, 95% CI [0.04, 0.09], p < 0.001 | r = 0.05, 95% CI [0.03, 0.06], p < 0.001 |
| F3 (bw) | r = −0.03, 95% CI [−0.07, 0.01], p = 0.170 | r = −0.01, 95% CI [−0.04, 0.02], p = 0.401 | r = −0.04, 95% CI [−0.02, −0.05], p < 0.001 | r = −0.02, 95% CI [−0.04, 0.01], p = 0.090 | r = −0.03, 95% CI [−0.01, −0.04], p < 0.001 |

See Table 1 for full description of abbreviated vocal characteristics. $n_{Study\ 1} = 56$, $n_{Study\ 2} = 277$.

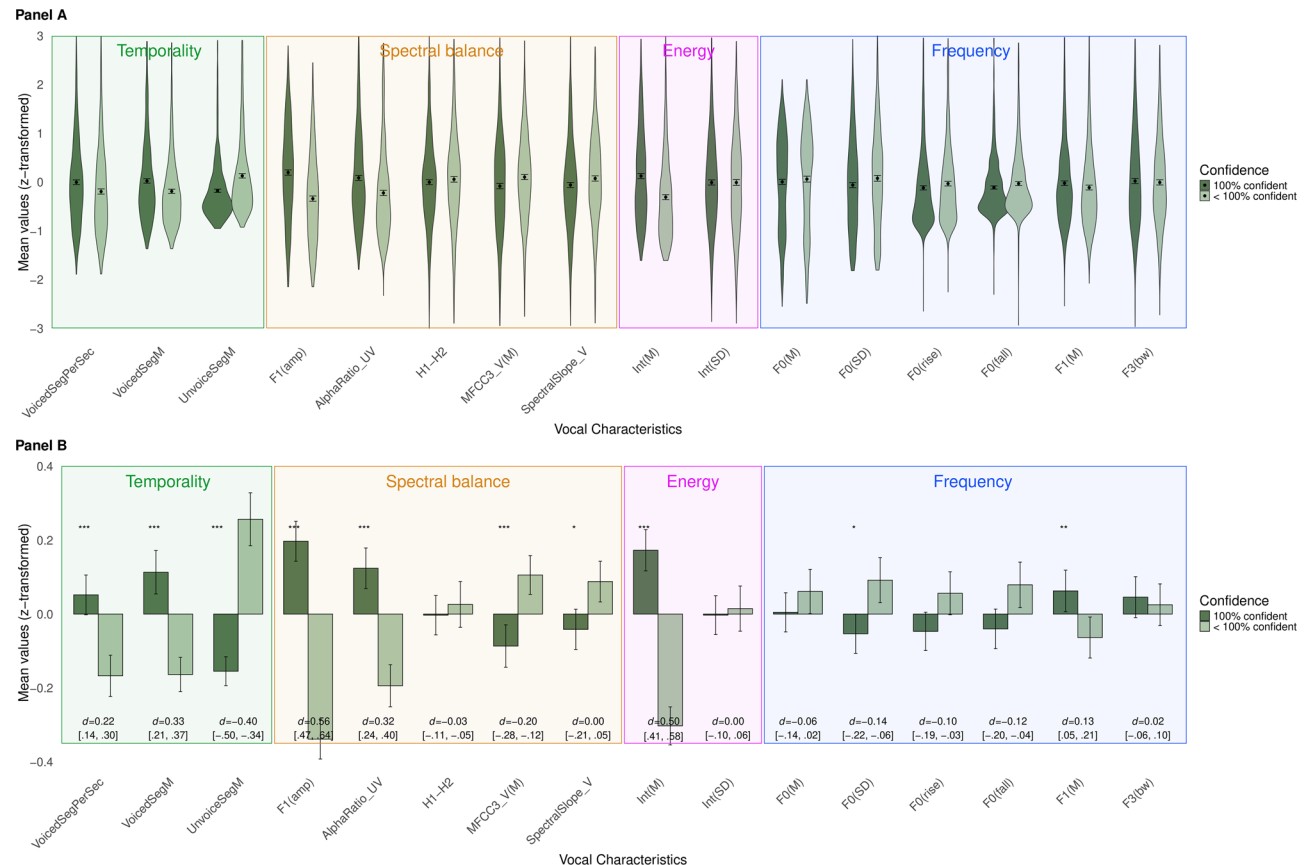

**Fig. 2 | Vocal Characteristics for Statements with Witness-Judged Confidence of 100% Or Less.** Legend: **A** displays violin plots showing the distribution of data, with mean and 95% confidence intervals. **B** zooms in on the means and 95% CIs, without showing the distribution, for a clearer display of the patterns of acoustic parameters across conditions. $n = 2428$. $d$ = Cohen's d, *$p < 0.05$, **$p < 0.01$, ***$p < 001$. Exact $p$-values are available in Table 2.

### Table. 4 | Mean (and Standard Deviation) Performance Scores across Samples

|        | d′          | Hit rate    | False alarm rate | C           |
|--------|-------------|-------------|------------------|-------------|
| Sweden | 0.26 (0.50) | 0.74 (0.15) | 0.66 (0.18)      | 0.63 (0.57) |
| USA    | 0.15 (0.32) | 0.61 (0.12) | 0.55 (0.14)      | 0.22 (0.36) |
| India  | 0.21 (0.32) | 0.67 (0.16) | 0.61 (0.15)      | 0.42 (0.47) |
| Total  | 0.19 (0.37) | 0.65 (0.15) | 0.59 (0.16)      | 0.36 (0.47) |

d′ = d prime, c = response bias, n = 277.

## Discussion

### Study 1 – discussion

We began by replicating previous results[2] in a sample almost twice as large. In particular, correct responses were uttered more quickly, with shorter pauses and more energy in the first-formant region (see Fig. 1). We also found large effects of greater amplitude and higher alpha ratio in unvoiced regions for correct statements. Thus, the current results strengthen the idea that non-verbal aspects of speech carry some information regarding the accuracy of the recalled information[18]. However, it should be noted that we did not find the expected effect of higher pitch for correct statements[2]. This suggests that pitch might not be a reliable cue for accuracy—which aligns with the contrasting finding regarding the relationship between pitch and confidence[4,22]—and highlights the importance of using big samples during replication attempts. In spite of the failed replication for this vocal characteristic, the results generally corroborate the idea that greater ease of retrieval indicates accurate recall[9,11], given that higher speech rate and shorter pauses can be considered consequences of a fluent, effortless retrieval.

The results also partially replicated previous findings regarding the relationship between confidence and vocal characteristics, with more confident speech having a higher speech rate, shorter pauses, and greater amplitude. This replicates previous studies[3–5,19–22]. Further, it indirectly demonstrates the tight association between confidence and accuracy[10,14–17], given that the patterns of vocal characteristics for accuracy and confidence are nearly identical (see Figs. 1 and 2). However, not all predictions were supported, as we found no evidence for a greater amplitude variability (cf.[4,22]) nor for pitch variability (cf.[4,20]). A possible explanation could be that our study analyzed natural, spontaneous speech from eyewitness testimony, whereas the previous studies utilized voice samples from actors who were instructed to use pseudo words[20], a specific speaking style[4,22], and/or specific sentences[4]. That said, as discussed above, we did replicate several other effects from studies that had used acted speech. Overall, we find that higher speech rate, shorter pauses and greater amplitude are the most robust vocal characteristics for confident speech.

### Study 2 – discussion

Extending research on the accuracy of judging transcriptions and video recordings of eyewitness testimony[37,38], we examined audio-only cues and compared performance across three culturally diverse groups. Hypotheses were largely supported. All groups judged the accuracy of eyewitness statements at above-chance levels and assigned higher confidence ratings to correct vs. incorrect statements. Performance was higher with greater cultural proximity to the stimuli (i.e., Swedish-spoken testimony), in descending order for Swedish, US, and Indian samples. Although significant, performance was modest, ranging around $d' = 0.15$–$0.21$. Accurately judging eyewitness testimony based on audio alone appears to be a difficult task, but we show that it is still possible. Similarities across samples suggest there

may be universally perceived vocal cues to accuracy, with *mean amplitude* the strongest vocal cue.

Surprisingly, we did not find that the participants' performance differed significantly between the three cultural groups, with moderate-to-anecdotal evidence supporting the null (BF = 0.23–0.75). This could possibly be due to the difficulty of the stimuli material, as many of the uttered statements were short in length and sometimes contained only one or a few words or syllables. This makes it more difficult to extract information from the statement, such as prosody (as well as semantic meaning for the Swedish participants). Future studies should examine cultural differences in testimony speech with longer utterances.

### General discussion

The main aim of this study was twofold: (1) to replicate the findings that correct statements in eyewitness testimonies are uttered with different vocal characteristics than incorrect statements, and (2) to examine to what extent people can judge accuracy in eyewitness testimony based on speech alone. We also examined the relationship between vocal characteristics and confidence, including both the witnesses' own judgments and observer judgments of confidence. We concluded by analyzing the effect of language comprehension on the ability to judge accuracy. In line with previous findings, Study 1 demonstrated that correct statements in eyewitness testimony—and statements judged with higher confidence by the witnesses themselves—were uttered with a higher speech rate, fewer pauses, and a greater amplitude. Study 2 demonstrated that participants could judge accuracy from speech at (slightly) above chance level, with similar performance regardless of language comprehension. Participants also judged that correct statements were uttered with more confidence than incorrect statements. We detail highlights and limitations below.

**Are there vocal cues to accuracy?.** Our findings suggest that there appear indeed to exist reliable vocal cues to the accuracy of eyewitness testimony. By examining the overlap between Figure 4 in Gustafsson et al.[2] ($n_{statements}$ = 1884) and Fig. 1 in the current study ($n_{statements}$ = 3344), we find that witnesses' speech takes on different vocal properties depending on if the recalled detail is correct or incorrect. Specifically, correct statements appear to be uttered with a greater energy (Int [M]), greater energy in the first-formant region (F1 [amp]), higher speech rate (VoicedSegPerSec), and shorter pauses (UnvoicedSeg [M]). In addition to these effects that clearly replicate previous findings[2], Study 1 contained several other statistically significant effects for vocal characteristics as well.

Interestingly, the vocal characteristics that indicate accuracy overlapped almost completely with findings on witnesses' own self-reported confidence (see Figs. 1 and 2). That is, when witnesses expressed being 100% confident in their accuracy (as compared to less than 100% confident) they had relatively similar vocal-characteristic levels as correct vs. incorrect statements. There is a known confidence-accuracy relationship in eyewitness research, with witnesses being more confident in correct memories[16,17], and the overlap when considering vocal cues is considerable. Although we cannot derive the causal relationship between speech and confidence, the current results clearly demonstrate that there is some attribute in a memory that simultaneously makes people feel confident and makes them speak in a specific manner in terms of acoustical properties—a manner which appears to be "straight forward", as evidenced by a more fluent (fewer pauses), fast (higher speech rate), and loud (greater mean amplitude and greater amplitude in the first-formant region) speech. One could postulate that feeling confident makes one speak more fluently, quickly, and loudly, which are some aspects of speech that listeners consider when assessing a speaker's confidence[3,4,19]. However, it is reasonable to assume that there are attributes even *prior* to the initial confidence that influence the speech—specifically, the underlying memory *strength*. Memory strength is an abstract term to describe a quality of the memory, which mainly involves how well encoded and easily retrieved it is.

The ease with which a memory is retrieved has been shown to be a cue to confidence[39], and we suggest that an easily retrieved memory will be uttered more fluently, that is with fewer and shorter pauses, and as a consequence, a higher speech rate. Thus, the strength of the memory likely affects both the confidence and the utterance of that memory.

**Do people attune to the vocal cues to accuracy?.** We found that both native (Swedish) and non-native (American and Indian) listeners were able to judge the accuracy of witnesses' statements (in Swedish) at above-chance levels (see Table 4). Performance was not great, however, residing at around 55% accuracy and an overall *d'* score of 0.19 (SD = 0.37; see Table 4), where flipping a coin would be accurate at 50%. Participants also displayed a clear "truth bias," as evident from the high false alarm rate (see Table 4). Research finds a similar effect in lie detection, that is, a tendency for people to be better at detecting lies than truth[40]. This liberal bias obtained here might result from an availability heuristic, where being exposed in daily life mostly to correct statements makes individuals more inclined to believe another person's recollection. This, in turn, suggests that the many real-life cases of mistaken eyewitnesses[1] might not have a large influence on beliefs about eyewitness accuracy. Albeit with this bias, the results presented here indicate that listeners are tuning in to some aspects of the speech that allow them to gauge the accuracy of the respective statement.

We expected listeners in the Swedish sample to perform most accurately because they would also have access to the verbal content (despite being instructed to focus only on the nonverbal aspects of the speech), which meant they could judge accuracy by weighing that information, such as the likelihood of remembering a specific type of detail (e.g., a knife, or the color of the offender's shoes). Swedish speakers would also be familiar with language-specific nonverbal vocal patterns (i.e., speech prosody[25,26]), and also cultural specificity relating more generally to vocal expression[28]. However, results provided moderate-to-anecdotal evidence that there were no differences between the language comprehension groups, possibly indicating that they relied on similar vocal cues. Evidence suggests that participants did rely on vocal cues in their judgments; the highest correlations were observed for loudness (Int [M], $r_{all}$ = 0.30), energy in the first-formant region (F1 [amp], $r_{all}$ = 0.25), speech rate (VoicedSegPerSec, $r_{all}$ = 0.18) and pause length (UnvoicedSeg [M], $r_{all}$ = −0.22). We note that the strongest correlations were present for the vocal characteristics that best indicated accuracy, suggesting that participants *did* tune in to valid cues.

Future research should examine how eyewitness accuracy judgments can be improved, for example, by asking participants to focus on the vocal aspects most indicative of accuracy (such as loudness). Another relevant area for future research is to examine potential cut-offs in specific vocal-characteristic levels, whereby a statement becomes more (or less) likely to be correct. Attempts have been made to find such cut-offs for the cue *response latency*, but this has proven difficult[41,42]. By contrast, focusing on a number of pauses and verbal hedges (e.g., "maybe", "perhaps") appears to improve accuracy at judging eyewitness testimony statements[37].

As mentioned, judgment accuracy among the participants was modest at best, residing only slightly above the chance level overall. Although this does take away from the potential applicability of the findings (i.e., it might not be worth the time and investment to teach people about the vocal cues to accuracy if their performance will not increase by more than a few percentage points), it is important to highlight the context of the data. That is, participants listened to ten randomized testimony statements from ten different witnesses, without knowing anything about the context of the crime that had been witnessed, nor anything about the questions that had probed the respective statement. Moreover, the audio clips (i.e., testimony statements) were generally quite short, with many lasting less than a second. Nevertheless, despite these minimal conditions, there was

still a valid signal coming through, and participants found vocal cues to accuracy. Given their non-verbal nature, these cues may also allow for greater generalization and universality compared to verbal cues, such as verbal confidence statements. However, this remains an empirical question for future studies.

It is also difficult to say to what extent vocal cues influence judgments of accuracy in real cases, where individuals who hear testimony also have contextual information and pre-existing beliefs[43]. Indeed, people likely judge accuracy based on a plethora of factors, such as semantics, phrasing, body language, and possibly clothing and race as well. These are not necessarily valid cues, and here we show that the vocal cues, at least to a degree are valid and can be detected. Future studies can examine how big a role the vocal cues play, and in turn compare how big of a role they *should* play. Our findings should be examined further and replicated before laying down guidelines on how to evaluate eyewitness testimony in level of vocal characteristics.

## Limitations

As mentioned in the introduction, many factors are known to distort memory[6]. Once memories become distorted, potential cues to accuracy become less reliable[10], so it is unlikely that the current vocal characteristics that indicate accuracy (loudness, speech rate, pause length) will generalize well to such situations. However, the extent to which the vocal cues to accuracy lose reliability over time remains an empirical question. In the meantime, we suggest that these vocal cues be best used for testimony that was obtained shortly after witnessing an event, in line with recommendations regarding the validity of confidence judgments[17].

Not all hypotheses were supported. In Study 1, we did not find a higher pitch for correct statements, nor greater amplitude variation and greater pitch variability for highly confident statements. This suggests that these vocal characteristics are less stable predictors of accuracy and confidence, respectively, and that the focus of research for practical purposes should focus more on the robust predictors discussed above. In Study 2, we did not find that the Swedish sample had a better ability to detect accuracy in the vocal samples than the other cultural samples. However, additional research is needed to support the idea of universal interpretations of speech accuracy. Another important caveat is that the accuracy predictions were rather modest (see Table 4), suggesting that *verbal content* should receive greater emphasis relative to vocal characteristics.

It is also important to highlight that the testimonies used in the experiments came from participants that voluntarily watched a video of a simulated crime, in a lab environment. This will differ from many real-life instances, where a witness is not likely prepared for the event, might not have their focus or undistracted view of it, and might be in danger. This means that the current conditions are unlikely to generalize completely to real events. Nonetheless, this type of design is typical for studies of eyewitness accuracy due to its ability to create a laboratory environment, and studies that examining real witnesses show similar accuracy rates between lab and real-world events[44].

The audio of the testimony statements was not recorded with professional-grade microphones and in a room that was not acoustically optimized for speech recordings. As a consequence, the data contains some levels of noise. Although this suggests that even better performance could be obtained with greater audio quality, eyewitnesses are also unlikely to be interviewed in proper studio environments.

Finally, we highlight the variation in the data. It is clear that there is a lot of variation, both when looking at the validity of the vocal characteristics for the individual statements and when looking at participants' ability to judge accuracy (which in turn is the case both between participants and within participants [i.e., between statements]). This is to some extent inevitable; there will be some correct statements that are uttered in an "atypical" way for correct statements (e.g., a slower response speed), and vice versa for incorrect statements. This means that no method, including this one based on vocal cues, can ever be perfect. Nonetheless, this is an important limitation to highlight because it does limit the applicability of its use. That said,

this study adds to the picture about memory strength, accuracy, and confidence and gives us yet another look into how accuracy can take effect. Whereas previous studies have shown that it can take place in the form of (para)verbal expressions of effort in memory retrieval[9,12,45], and in witnesses' own confidence judgments[9,14,17], we show that there is something in a memory's accuracy (and/or memory strength) that is expressed nonverbally in the speech of that recalled memory.

## Conclusions

The current study strengthens the idea that memory accuracy is not just conveyed in *what* we say, but also *how* we say it. In particular, we studied the accuracy of eyewitness statements and judgments of their accuracy by untrained observers. We find that there are vocal cues to accurate memories, which extend and largely replicate past work. Moreover, observers are capable, to some extent of picking up on the accuracy of these eyewitness statements based on audio alone. Impressively, this ability is not just present for people who can comprehend the language but also for people that are unfamiliar with the language and thus solely have nonverbal vocal cues to inform them.

## Data availability

Data is available on the Open Science Framework (Study 1, https://osf.io/x7u5g; Study 2, https://osf.io/x7u5g). Data for Study 1 contain values for all vocal cues included in the analyses for all included witness statements. Data for Study 2 contain anonymized data from all participants in the human judgment study. The conditions of ethics approval and consent procedures do not permit the public archiving of the audio files containing the witness statements.

## Code availability

Code is available on the Open Science Framework (Study 1, https://osf.io/x7u5g; Study 2, https://osf.io/x7u5g). This code can be used to reproduce the analyses and figures, and requires the statistical software R[31], and the packages *lme4*[32], *tidyverse*[33] and *BayesFactor*[46].

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

## Acknowledgements

This research was supported by the Marianne and Marcus Wallenberg Foundation (MMW 2018.0059). The funders had no role in study design, data collection and analysis, decision to publish, or the preparation of the manuscript.

## Author contributions

P.U.G.: Conceptualization, Data curation, Formal analysis, Investigation, Methodology, Project administration, Resources, Visualization, Writing - original draft, and Writing - review & editing. P.L.: Conceptualization, Formal analysis, Funding acquisition, Methodology, Project administration, and Writing - review & editing. H.A.E.: Investigation, Methodology, and Writing - review & editing. N.S.T.: Investigation and Writing - review & editing.

## Funding

## Competing interests

The authors declare no competing interests.
