## [Transparent Peer Review file · Communications Psychology]

Vocal Cues to Eyewitness Accuracy are Detected by Listeners With and Without Language Comprehension

Corresponding Author: Dr Philip Gustafsson

Version 0:

Decision Letter:

Dear Dr Gustafsson,

Thank you for your patience during the peer-review process. Your manuscript titled "Eyewitness Testimony Accuracy Is Conveyed Through Vocal Cues with Both Cultural Universals and Differences" has now been seen by 2 reviewers with highly relevant expertise in eyewitness testimony and acoustic analyses of voices, whose comments are appended below. You will see that they find your work of potential interest. However, they have raised substantial concerns that must be addressed. In light of these comments, we cannot accept the manuscript for publication, but would be interested in considering a revised version that fully addresses these concerns.

We hope you will find the Reviewers' comments useful as you decide how to proceed. Should additional work allow you to address these criticisms, we would be happy to look at a revised manuscript. If you choose to take up this option, please highlight all changes in the manuscript text file, and provide a detailed point-by-point reply to the reviewers.

Editorially, we consider it important that the revised manuscript provide appropriate statistics to test the hypotheses. All preregistered analyses should be reported as well as additional tests that can fully address the research questions and hypotheses. Please also justify correction for multiple comparisons. In the case of null results, please do not interpret findings derived from null-hypothesis significance testing or ambiguous results (e.g., Bayes Factors $>.03$ and <3). To interpret null results please add Bayes Factor or Equivalence Tests. In regard to your power analyses, please add justifications for why you chose to power to $d = .02$ (study 1) and $d = .25$ (study 2). We also ask that you plot performance for each group (accuracy/confidence) in addition to reporting the derived statistics (z-values). Please ensure you follow our statistical guidelines when reporting statistics (<https://www.nature.com/commspsychol/submit/submission-guidelines#statistical-guidelines>).

I am attaching a checklist that details critical reporting requirements for the revised manuscript. Please attend to each item and ensure your manuscript is fully compliant. We are requesting that your manuscript aligns with these requirements as this facilitates the evaluation of your manuscript, reducing delays in re-review and potential future acceptance. If your revised manuscript is not aligned with these requests on major issues, such as those concerning statistics, it may be returned to you for further revisions without re-review. Additional information can be found in our style and formatting guide [a href="https://www.nature.com/documents/commspsychol-style-formatting-guide-accept.pdf">Communications Psychology formatting guide](https://www.nature.com/documents/commspsychol-style-formatting-guide-accept.pdf).

If the revision process takes significantly longer than five months, we will be happy to reconsider your paper at a later date, provided it still presents a significant contribution to the literature at that stage.

Please use the following link to submit your

- revised manuscript,
- point-by-point response to the referees' comments,
- cover letter (as a separate document),
- the Editorial Policy Checklist (see below),
- the Reporting Summary (see below), and
- the completed Editorial Request Table (attached):

Link Redacted

Thank you for the opportunity to review your work.

Best regards,

Saloni Krishnan

Saloni Krishnan, PhD
Editorial Board Member
Communications Psychology
orcid.org/0000-0002-6466-141X

REVIEWER EXPERTISE:

Reviewer #1: Eyewitness testimony, Acoustic analysis of voices

Reviewer #2: Eyewitness testimony

REVIEWER REPORTS:

Reviewer #1 (Remarks to the Author):

GENERAL COMMENTS

The authors indicated two goals for the study: In Study 1, the goal was to replicate and extend previous research examining the relationship of selected vocal/acoustic cues to accuracy and confidence of eyewitness testimony as indicated by the witness' accuracy and judged confidence in their decisions. In Study 1, three of four hypotheses for accurate statements compared to inaccurate statements were supported (for accurate statements, greater energy in the first formant region, higher speech rate, and shorter pauses were observed). The predicted effect of higher pitch in accurate statements was absent. For judged confidence by witnesses in their eyewitness judgments, support for three of the five hypotheses was obtained. Utterances judged as "maximum confidence" exhibited a higher speech rate, shorter pauses, and greater amplitude. The predicted effects of greater amplitude variation and greater pitch variability in "maximum confidence" utterances were not obtained. In addition, for confidence, exploratory findings indicated that "maximum confidence" utterances exhibited lower pitch, higher pitch and greater energy in the first formant region, greater energy in higher frequencies of the spectrum for unvoiced sounds, lower mean of the third Mel-Frequency Cepstral Coefficient, for voiced regions, lower spectral slope in the 500–1500 Hz region for voiced areas, and longer-lasting speech.

In Study 2, there were two goals. The first goal was to examine observers' judgments of accuracy and confidence for the eyewitness statements, to see if their judgments matched the judgments of the eyewitnesses themselves. The second goal was to examine multicultural factors by comparing three cultural groups: Swedish, American, and Indian. For accuracy, the predictions were that observers' judgments of accuracy would be above chance with judgments most accurate for Swedish speakers followed by Americans and Indians. For confidence, the authors predicted that confidence judgments would be higher for correct statements, highest for Swedish speakers followed by Americans and Indians, and would be correlated between eyewitnesses and observers, with the degree of correlation highest for Swedish speakers followed by Americans and Indians. The results showed that observers' accuracy judgments were above chance, with no difference across the three cultural groups. This finding did not support their hypothesis. For confidence, judgments were highest for Indians followed by Swedish and Americans. This did not support the hypothesis. However, the authors claimed that an interaction between country and the difference between correct and incorrect judgments fit the pattern. There was a correlation between eyewitness' and observers' judgments of confidence for the Swedish and American samples, but not the Indian sample. However, based on planned comparisons and an interaction between accuracy and group, the authors claimed the results fit their predictions across cultural groups, with the correlation between eyewitness and observer judgments of confidence greatest for the Swedish speakers followed by Americans and Indians.

Overall, this is a well-intentioned attempt to further examine the vocal/acoustic factors that affect eyewitness' verbal recollections of a witnessed event. In addition, the comparison of different cultures is worthwhile and addresses the generality of judgments across cultures. For both studies, the statistical analyses are complicated and time consuming to unpack, but they seem appropriate. In Study 1, six of nine predictions were supported regarding accuracy and confidence. One issue I have is that the absence of support for three predictions was not addressed or explained either in the Study 1

Discussion or General Discussion. Do the authors have any possible explanations for the failure to support those predictions?

In Study 2, we have the same issue again, which is the absence of explanation for hypotheses that were not supported. For example, do the authors have an explanation for why there was no overall difference in accuracy judgments across the three cultural groups? I also have an issue with the interpretation of the interactions between country and other variables related to the confidence judgments (lines 361-368; lines 372-378). I am not sure that the DIFFERENCE between correct and incorrect statements is the appropriate metric instead of just relying on the overall main effect of country. And in terms of comparing the magnitude of Pearson-R numbers and claiming that the numbers reflect a difference across groups, to the tune of 4.5 times greater, 13.5 times greater, and 3.0 times greater, it is my understanding this kind of comparison is prohibited. There are many reasons different correlation values are obtained in experiments and they do mean anything in terms of magnitudes of difference across groups.

I found the General Discussion to be brief. I would like to see more discussion of the failures of supporting certain hypotheses and what the impact of what those failures are, if any. I would also like to see more discussion of the practical ramifications of their findings on judgments of observers who listen to the eyewitness statements. This appears to be the critical main thrust of the article. And finally, I wonder if the authors can identify any vocal/structural differences across the three languages that could have affected the results. I know that some tonal languages like Mandarin would be very different than English in terms of how vocal aspects of utterances are processed. This was mentioned in the Introduction (lines 102-104) but not addressed later.

SPECIFIC COMMENTS

line 43 This is a general statement that needs some clarification

line 126 What was this previous research, possibly include citations.

Lines 313-317 I wonder if giving the American and Indian samples this instruction created a confound. Would participants normally focus their attention on these aspects of the utterances in their own languages?

Line 337 Table 4 – The false alarm rates are very high, do the authors have any explanation for this outside just saying it's a liberal bias?

Reviewer #2 (Remarks to the Author):

I reviewed the manuscript entitled "Eyewitness Testimony Accuracy Is Conveyed Through Vocal Cues with Both Cultural Universals and Differences". After reading the abstract, I was excited to read this manuscript. While there has been much work conducted on the subject of eyewitness identification confidence and factors that relate to eyewitness identification accuracy, much less work has focused on witness testimonial statements. The intersection of speech research and eyewitness testimony seems an interesting and understudied area. This area of research has the potential to make both a novel and important contribution to the legal system. However, upon reading the manuscript, several key concerns emerged. Many of these concerns arose from the length of the manuscript. While I appreciate the authors' attempt to keep the manuscript concise and understand journal formatting guidelines, the brevity in the manuscript caused problems in each section.

The introduction of the manuscript felt very jumbled. It was difficult to find a streamlined narrative that justified the current studies. Rather, it felt that topics jumped around frequently and thus, by the end of the introduction, the justification for the current studies felt weak and some of the past literature was discussed in an overly broad manner. For instance, in discussing eyewitness confidence, the authors cite the Wixted and Wells (2017) review paper. However, this paper focuses on eyewitness identification confidence and not eyewitness testimonial confidence. Moreover, the relationship between confidence and accuracy discussed in that paper is not absolute, but rather depends on the circumstances of the lineup. The authors make claims related to confidence like "nonverbal cues for confidence levels can vary across cultures and reduce cross-cultural accuracy" but do not provide support for this claim. The authors state that "speaker confidence is a potentially valid cue from speech research that is worth expanding to study in the eyewitness context". I would have liked to see them expand on this statement to really bring home the point about why this research is needed.

In Study 1, the authors measure various aspects of vocal characteristics of eyewitness testimonial statements. However, the authors do not explain what these characteristics are or why they are chosen. For instance, what is the first formant region and why are the authors predicting this will be associated with correct statements. Speech researchers might be familiar with these vocal cues but memory researchers or criminal justice researchers likely will not. I also found it concerning that the authors placed the effect sizes in the supplemental materials as I think this is a needed statistic in the main manuscript (in Study 2, effect sizes are also missing for several analyses). The discussion section for study 1 is also very short and does not effectively communicate what the contribution of this study is beyond replicating past work. If the only goal of this study was to replicate past work, more detail is needed about why this replication was necessary and how this replication adds to existing knowledge. Additionally, the authors state that because the patterns of vocal characteristics for accuracy and confidence are similar, this means there is an association between confidence and accuracy. I do not really follow the logic here. Just because vocal characteristics for accuracy and confidence are similar does not necessarily mean that confidence is related to accuracy. Study 2 was by far more compelling in that it addressed a novel research question and seemed better

supported by the introduction.

The discussion of the manuscript was also quite short and did not discuss the practical implications of this work despite situating the study as applied in the abstract and introduction. The authors discuss that even though their design was conservative and there was no contextual information, participants were still able to judge accuracy. However, this is the opposite of the applied context the researchers are studying. In real cases, those judging testimonial accuracy will have contextual information and pre-existing beliefs. How would these results change in those circumstances? Are the vocal cues the authors studying able to be identified by listeners in everyday conversations and, if not, what do the authors think about how these results could be applied in a legal context? Answering questions like these would have strengthened the discussion.

EDITORIAL POLICIES

We ask that you ensure your manuscript complies with our editorial policies and reporting requirements.

To that end, we require revised manuscripts to be accompanied by two completed items: a reporting summary that collects information on study design and procedure, and an editorial policy checklist that verifies compliance with all required editorial policies

- <https://www.nature.com/documents/nr-reporting-summary.zip>>Nature Research Reporting Summary
- <https://www.nature.com/documents/nr-editorial-policy-checklist.pdf>>Editorial Policy Checklist

All points on the policy checklist must be addressed. Your revised manuscript can only be sent back to the referees if these checklists are completed and uploaded with the revision.

Notes: If you have submitted a Stage 1 Registered Report, Review, Primer, Comment, or Perspective you do not need to submit these forms. If you have already submitted these forms, you may disregard this request.

** Visit Nature Research's author and referees' website at <http://www.nature.com/authors>>www.nature.com/authors for information about policies, services and author benefits**

If you experience problems in linking your ORCID, please contact the <http://platformsupport.nature.com/>>Platform Support Helpdesk.

Version 1:

Decision Letter:

Dear Dr Gustafsson,

Thank you for submitting your manuscript titled "Eyewitness Testimony Accuracy Is Conveyed Through Vocal Cues with

Both Cultural Universals and Differences" to Communications Psychology. We have given the paper our careful consideration and find it of potential interest. However, due to certain shortcomings we are concerned that sending the current manuscript out to review could lead to unnecessary delays and quite possibly an undesirable outcome of the review process.

In particular, we would like you to address the points we raised in our previous decision letter about the statistics reporting which includes reporting confidence intervals around all effect sizes. We require that every results statement be supported by appropriate inferential statistics. Please also ensure that you submit the Supplementary Information that provides the requested summary of the data.

We would therefore like to invite you to revise your manuscript to address these concerns before we make a final determination on whether to send your manuscript for external review.

We shall hope to receive your revised version as soon as you are able to complete the suggested revisions. If something similar is published in the interim we will have to consider the impact it has on the novelty of a revised manuscript.

If you anticipate a delay of more than four weeks, please let us know. Should your manuscript be substantially delayed without notifying us in advance and your article is eventually published, the received date may be that of the revised, not the original, version.

We also ask that you ensure your manuscript complies with our editorial policies and reporting requirements.

To that end, we require revised manuscripts to be accompanied by two completed items: a reporting summary that collects information on study design and procedure, and an editorial policy checklist that verifies compliance with all required editorial policies.

- <https://www.nature.com/documents/nr-reporting-summary.zip>>Nature Research Reporting Summary
- <https://www.nature.com/documents/nr-editorial-policy-checklist.pdf>>Editorial Policy Checklist

All points on the policy checklist must be addressed. Your revised manuscript can only be sent to referees if these checklists are completed and uploaded with the revision.

If you are not interested in submitting a suitably revised manuscript in the future please let me know immediately so we can close your file. If you have any questions, please contact me.

Please use the link below when you are prepared to resubmit.
Link Redacted

Thank you for your interest in Communications Psychology.

Best regards,
Saloni Krishnan

Saloni Krishnan, PhD
Editorial Board Member
Communications Psychology
orcid.org/0000-0002-6466-141X

Version 2:

Decision Letter:

Dear Dr Gustafsson,

Your manuscript titled "Eyewitness Testimony Accuracy Is Conveyed Through Vocal Cues with Both Cultural Universals and Differences" has now been seen by our reviewers, whose comments appear below. In light of their advice I am delighted to say that we are happy, in principle, to publish a suitably revised version in Communications Psychology.

We therefore invite you to revise your paper one last time to address the remaining concerns of our reviewers and a list of editorial requests. At the same time we ask that you edit your manuscript to comply with our format requirements and to maximise the accessibility and therefore the impact of your work.

EDITORIAL REQUESTS:

SUBMISSION INFORMATION:

OPEN ACCESS:

* DATA AVAILABILITY:

Link Redacted

Best regards,

Jennifer Bellingtier

Jennifer Bellingtier, PhD
Senior Editor
Communications Psychology

Saloni Krishnan, PhD
Editorial Board Member
Communications Psychology
orcid.org/0000-0002-6466-141X

REVIEWERS' EXPERTISE:

Reviewer #1 Eyewitness testimony, Acoustic analysis of voices

REVIEWERS' COMMENTS:

Reviewer #1 (Remarks to the Author):

The authors have completed a good faith revision of the manuscript. I focused primarily on my concerns and I find that the authors have satisfactorily addressed my comments. As well, I believe the manuscript is much stronger with the revisions predicated on both reviewers' comments, additional pertinent references, and the addition of other statistical tests. I have no other comments on the paper to add.

Reviewer comments: *Eyewitness Testimony Accuracy Is Conveyed Through Vocal Cues with Both Cultural Universals and Differences*

We appreciate this opportunity to resubmit a revised version of our manuscript and the helpful feedback that you have provided. Below the original feedback appears in black font and our response in blue font. Please note that italicized texts refer to texts in the manuscript.

Reviewer #1:

Overall, this is a well-intentioned attempt to further examine the vocal/acoustic factors that affect eyewitness' verbal recollections of a witnessed event. In addition, the comparison of different cultures is worthwhile and addresses the generality of judgments across cultures. For both studies, the statistical analyses are complicated and time consuming to unpack, but they seem appropriate.

1. One issue I have is that the absence of support for three predictions was not addressed or explained either in the Study 1 Discussion or General Discussion. Do the authors have any possible explanations for the failure to support those predictions?

We have now expanded the discussion in Study 1, which now also includes a discussion about failed predictions:

p. 18, lines 14-19:

However, it should be noted that we did not find the expected effect of higher pitch for correct statements, as originally found in Gustafsson et al. (2023). This suggests that pitch might not be a reliable cue for accuracy—which aligns with the contrasting finding regarding the relationship between pitch and confidence (e.g., Jiang & Pell, 2017; Van Zandt & Berger, 2020)—and highlights the importance of using big samples during replication attempts.

pp. 18-19, lines 32-41:

However, not all predictions were supported, as we found no evidence for a greater amplitude variability (cf. Jiang & Pell, 2017; Van Zandt & Berger, 2020) nor for pitch variability (cf. Goupil et al., 2021; Jiang & Pell, 2017). A possible explanation could be that our study analyzed natural, spontaneous speech from eyewitness testimony, whereas the previous studies utilized voice samples from actors who were instructed to use pseudo words (Goupil et al., 2021), a specific speaking style (Jiang & Pell, 2017; Van Zandt & Berger, 2020), and/or specific sentences (Jiang & Pell, 2017). That said, as discussed above we did replicate several other effects from studies that had used acted speech. Overall we find that higher speech rate, shorter pauses and greater amplitude are the most robust vocal characteristics for confident speech.

2. In Study 2, we have the same issue again, which is the absence of explanation for hypotheses that were not supported. For example, do the authors have an explanation for why there was no overall difference in accuracy judgments across the three cultural groups?

We have expanded the discussion in Study 2 as well, which now also includes a discussion about failed predictions:

pp. 25-26, lines 199-204:

Surprisingly, we did not find that the participants performance differed significantly between the three cultural groups. This could possibly be due to the difficulty of the stimuli material, as many of the uttered statements were short in length and sometimes contained only one or a few words or syllables. This makes it more difficult to extract information from the statement, such as prosody (as well as semantic meaning for the Swedish participants). Future studies should examine cultural differences in testimony speech with longer utterances.

3. I also have an issue with the interpretation of the interactions between country and other variables related to the confidence judgments (lines 361-368; lines 372-378). I am not sure that the DIFFERENCE between correct and incorrect statements is the appropriate metric instead of just relying on the overall main effect of country. And in terms of comparing the magnitude of Pearson-R numbers and claiming that the numbers reflect a difference across groups, to the tune of 4.5 times greater, 13.5 times greater, and 3.0 times greater, it is my understanding this kind of comparison is prohibited. There are many reasons different correlation values are obtained in experiments and they do mean anything in terms of magnitudes of difference across groups.

We appreciate this opportunity to clarify an area where the previous version of the manuscript was unclear. The interaction term of accuracy x country on confidence indicates the extent to which participants would perceive correct statements as being uttered with higher confidence compared to incorrect statements, which is Hypothesis 2a. Relying on the main effect for country would provide information as to whether one of the cultures generally perceived *all* statements to have a greater level of confidence relative to how the other cultures perceived it, which is a very interesting different question.

To address this important point regarding coefficient comparisons, we now follow Zou (2007) and have calculated confidence intervals in order to allow comparisons of differences between the groups:

pp. 24-25, lines 176-180:

Using Pearson correlations, we found statistically significant correlations for the Swedish sample, $r = .189$, 95% CI [.162, .216], $p < .001$; and the American sample $r = .042$, 95% CI [.025, .059] $p < .001$; but not for the Indian sample, $r = .014$, 95% CI [-.014, .042] $p = .399$ (see Table 3), supporting H3a. Overall, there was a statistically significant correlation across groups, $r = .069$, $p < .001$.

4. I found the General Discussion to be brief. I would like to see more discussion of the failures of supporting certain hypotheses and what the impact of what those failures are, if any. I would also like to see more discussion of the practical ramifications of their findings on judgments of observers who listen to the eyewitness statements. This appears to be the critical main thrust of the article. And finally, I wonder if the authors can identify any vocal/structural differences across the three languages that could have affected the results. I know that some tonal languages like Mandarin would be very different than English in terms of how vocal aspects of utterances are processed. This was mentioned in the Introduction

(lines 102-104) but not addressed later.

We now discuss failed predictions in the general discussion, including practical ramifications:

p. 31, lines 325-334:

Not all hypotheses were supported. In Study 1, we did not find higher pitch for correct statements, nor greater amplitude variation and greater pitch variability for highly confident statements. This suggests that these vocal characteristics are less stable predictors of accuracy and confidence respectively, and that the focus of research for practical purposes should focus more on the robust predictors discussed above. In Study 2, we did not find that the Swedish sample had a better ability to detect accuracy in the vocal samples than the other cultural samples. If this finding can be replicated, it would suggest universal interpretations of speech accuracy. An important caveat however is that the accuracy predictions were rather modest (see Table 4), suggesting that verbal content should receive greater emphasis relative to vocal characteristics.

Regarding vocal/structural differences, we did not find any clear examples of culture-specific correlations between perceived confidence and vocal cues and now explicitly mention this in our discussion (“Do people attune to the vocal cues to accuracy?”):

pp. 28-29, lines 268-284:

We expected listeners in the Swedish sample to perform most accurately because they would also have access to the verbal content (despite being instructed to focus only on the nonverbal aspects of the speech), which meant they could judge accuracy by weighing that information, such as the likelihood of remembering a specific type of detail (e.g., a knife, or the color of the offender’s shoes). Swedish speakers would also be familiar with language-specific nonverbal vocal patterns (i.e., speech prosody; e.g., Fodor, 1998; Wagner & Watson, 2010), and also cultural specificity relating more generally to vocal expression (e.g., Laukka & Elfenbein, 2021). However, results showed no major differences between the language comprehension groups, possibly indicating that they relied on similar vocal cues. Indeed, such an idea is strengthened when examining the correlation between listener-perceived confidence in the witnesses’ speech, and the vocal characteristics of that speech. All language-comprehension groups had similar patterns of correlation coefficients, without clear cultural effects. The highest correlations were observed for loudness (Int [M], rall = .30), energy in the first-formant region (F1 [amp], rall = .25), speech rate (VoicedSegPerSec, rall = .18) and pause length (UnvoicedSeg [M], rall = -.22). We note that the strongest correlations were present for the vocal characteristics that best indicated accuracy, suggesting that participants did tune in to valid cues.

SPECIFIC COMMENTS

5. line 43 This is a general statement that needs some clarification

We have now rephrased the sentence to be clearer:

p. 3, lines 41-43:

There are many known cases whereby eyewitnesses have been wrong in their recall or identification, which has led to innocent convictions (Innocence project, n.d.), making research on witness accuracy important for social justice.

6. line 126 What was this previous research, possibly include citations.

We have now added references to this in the introduction:

p. 4, lines 83-88:

Regarding pitch, findings have been inconsistent, with some studies finding that confident speech is uttered with higher pitch (see Van Zandt & Berger, 2020; Scherer et al., 1973) and others that confident speech is uttered with lower pitch (Apple et al., 1979; Guyer et al., 2019; Jiang & Pell, 2017).

7. Lines 313-317 I wonder if giving the American and Indian samples this instruction created a confound. Would participants normally focus their attention on these aspects of the utterances in their own languages?

We now clarify that there is no confound because all three groups received the same instructions to focus on the nonverbal aspects of the voice and not on the meaning of the words when they made their judgment.

p. 22, lines 115-119:

Furthermore, all participants were informed that they should focus on the “nonverbal aspects of the speech, such as the tone of voice” rather than on the verbal content when making their judgments. The American and Indian participants were additionally informed that the statements were spoken in Swedish and they were not expected to understand the content of the speech.

8. Line 337 Table 4 – The false alarm rates are very high, do the authors have any explanation for this outside just saying it’s a liberal bias?

We have now added a bit about this in the general discussion:

p. 28, lines 258-265:

Participants also displayed a clear “truth bias,” as evident from the high false alarm rate (see Table 4). Research finds a similar effect in lie detection, that is, a tendency for people to be better at detecting lies than truth (see e.g., Vrij & Baxter, 1999). This liberal bias obtained here might result from an availability heuristic, where being exposed in daily life to mostly to correct statements makes individuals more inclined to believe another person’s recollection. This in turn suggests that the many real-life cases of mistaken eyewitnesses (see Innocenceproject, n.d.) might not have a large influence on beliefs about eyewitness accuracy.

Reviewer #2:

9. While I appreciate the authors’ attempt to keep the manuscript concise and understand journal formatting guidelines, the brevity in the manuscript caused problems in each section.

We have now expanded each section of the manuscript to increase the narrative, depth and clarity of the text.

10. The introduction of the manuscript felt very jumbled. It was difficult to find a streamlined narrative that justified the current studies. Rather, it felt that topics jumped around frequently and thus, by the end of the introduction, the justification for the current studies felt weak and some of the past literature was discussed in an overly broad manner. For instance, in discussing eyewitness confidence, the authors cite the Wixted and Wells (2017) review paper. However, this paper focuses on eyewitness identification confidence and not eyewitness testimonial confidence. Moreover, the relationship between confidence and accuracy discussed in that paper is not absolute, but rather depends on the circumstances of the lineup. The authors make claims related to confidence like “nonverbal cues for confidence levels can vary across cultures and reduce cross-cultural accuracy” but do not provide support for this claim. The authors state that “speaker confidence is a potentially valid cue from speech research that is worth expanding to study in the eyewitness context”. I would have liked to see them expand on this statement to really bring home the point about why this research is needed.

We have now rewritten the introduction with an attempt to build a stronger narrative, including more specific effects in the literature. For example, regarding confidence, we have now added both references to eyewitness identifications and testimony alike (as they both show same trend of effects), but added that correct memories are not always rated with higher confidence:

p. 4, lines 65-68:

Moreover, the witness generally rates themselves as more confident when their memories are correct (albeit not always), both eyewitness testimony and line-up identifications (e.g., Gustafsson et al., 2022; Juslin et al., 1996; Robinson et al., 1997; Spearing & Wade, 2022; Wixted & Wells, 2017).

We have also expanded the introduction to more strongly motivate research on the relationship between vocal characteristics of confident speech in eyewitness testimony:

pp. 4-5, lines 75-91:

Although there are few studies on vocal characteristics and eyewitness accuracy (Goupil & Aucouturier, 2021; Gustafsson et al., 2023), considerable evidence exists on the relationship between vocal characteristics and confidence more generally. Specifically, studies find that speech is judged (and produced) as more persuasive and confident when uttered with a higher speech rate (including shorter pauses; Apple et al., 1979; Goupil et al., 2021; Guyer et al., 2019; Kimble & Seidel, 1991; Scherer et al., 1979, Van Zandt & Berger, 2020), greater amplitude (Jiang & Pell, 2017; Kimble & Seidel, 1991; Scherer et al., 1979, Van Zandt & Berger, 2020) and a greater amplitude variation (Jiang & Pell, 2017; Van Zandt & Berger, 2020). Regarding pitch, findings have been inconsistent, with some studies finding that confident speech is uttered with higher pitch (see Van Zandt & Berger, 2020; Scherer et al., 1973) and others that confident speech is uttered with lower pitch (Apple et al., 1979; Guyer et al., 2019; Jiang & Pell, 2017). It is worthwhile to examine vocal characteristics and confidence in an eyewitness testimony setting, to determine to what extent these results generalize to a forensic setting.

This paper asks two research questions: First, do speakers use different vocal cues depending on their level of confidence? Second, do listeners perceive these potential vocal cues to confidence?

11. In Study 1, the authors measure various aspects of vocal characteristics of eyewitness testimonial statements. However, the authors do not explain what these characteristics are or why they are chosen. For instance, what is the first formant region and why are the authors predicting this will be associated with correct statements. Speech researchers might be familiar with these vocal cues but memory researchers or criminal justice researchers likely will not.

Thank you for highlighting this. We understand that many of the acoustic variables are rather technical and difficult to unpack. Indeed, many of these vocal characteristics are difficult to explain in layman terms while retaining technical accuracy. To mitigate this, we categorize them into four overarching groups, that *can* generally be explained in more easily understood terms. These terms are *frequency, energy, spectral balance, and temporal characteristics*. We have clarified this categorization of the acoustic variables, and added descriptions of them:

p. 8, lines 170-177:

The audio files were analyzed along the 16 acoustic dimensions used in Gustafsson et al. (2023). Gustafsson et al. (2023) chose these dimensions by running a principal-component analysis on 88 acoustic dimensions, as well as by extrapolating findings from confidence speech research. These acoustic dimensions represent different aspects of frequency (e.g., fundamental frequency which is related to perceived pitch), energy (related to perceived loudness), spectral balance (related to perceived voice quality), and temporal characteristics (e.g., speech rate and pauses) of the voice signal. Table 1 shows a description of each acoustic dimension (i.e., vocal characteristic).

Next, why might F1 (amp) – that is, relative energy of the spectral envelope in the first formant region – be related to correct statements? The straight-forward explanation is that it's backed by data: our previous study that explored relations between accuracy and vocal characteristics, found such a relationship. This is now more clearly outlined in the manuscript:

p. 4, lines 68-74:

Recently, Gustafsson et al. (2023) demonstrated that there are also nonverbal vocal cues to accuracy; correct statements in eyewitness testimony were uttered with higher pitch, greater energy in the first formant region, higher speech rate and shorter pauses (see also Goupil & Aucouturier, 2021). However, more studies are needed to assess the reliability of these initial findings. Moreover, existing research has not yet examined whether people can detect that these are vocal cues to accuracy.

Moreover, one could tentatively reflect that this relationship might occur because correct memories should theoretically have stronger memory traces (meaning a clearer image of the memory), and strong memories should be easier to verbalize than weak memories. This stronger verbalization could manifest as a more loudly produced utterance, and F1 (amp) is indeed a form of *Energy* acoustic variable. Moreover, well-defined, clear vowel sound are

linked to the first-formant spectrum, which may explain why this effect is found in this specific spectrum.

12. I also found it concerning that the authors placed the effect sizes in the supplemental materials as I think this is a needed statistic in the main manuscript (in Study 2, effect sizes are also missing for several analyses).

We have now added effect sizes to the figures, and also added effect sizes to the one-sample t-tests in Study 2:

pp. 22-23, lines 131-136:

We tested this with one-sample t-tests against 0 ($d' = 0$ indicates chance performance). In line with predictions (H1a), all groups performed above chance level; Swedish sample: $t(60) = 3.98, p < .001, \text{Cohen's } d = 0.51$; American sample: $t(155) = 5.94, p < .001, \text{Cohen's } d = 0.48$; Indian sample: $t(59) = 5.09, p < .001, \text{Cohen's } d = 0.66$ (see Table 4), for an overall score across groups above chance level: $t(276) = 8.47, p < .001, \text{Cohen's } d = 0.51$.

13. The discussion section for study 1 is also very short and does not effectively communicate what the contribution of this study is beyond replicating past work. If the only goal of this study was to replicate past work, more detail is needed about why this replication was necessary and how this replication adds to existing knowledge.

The motivation for Study 1 has now more clearly been added to the introduction:

pp. 4-5, lines 68-91:

Recently, Gustafsson et al. (2023) demonstrated that there are also nonverbal vocal cues to accuracy; correct statements in eyewitness testimony were uttered with higher pitch, greater energy in the first formant region, higher speech rate and shorter pauses (see also Goupil & Aucouturier, 2021). However, more studies are needed to assess the reliability of these initial findings. Moreover, existing research has not yet examined whether people can detect that these are vocal cues to accuracy.

Although there are few studies on vocal characteristics and eyewitness accuracy (Goupil & Aucouturier, 2021; Gustafsson et al., 2023), considerable evidence exists on the relationship between vocal characteristics and confidence more generally. Specifically, studies find that speech is judged (and produced) as more persuasive and confident when uttered with a higher speech rate (including shorter pauses; Apple et al., 1979; Goupil et al., 2021; Guyer et al., 2019; Kimble & Seidel, 1991; Scherer et al., 1979, Van Zandt & Berger, 2020), greater amplitude (Jiang & Pell, 2017; Kimble & Seidel, 1991; Scherer et al., 1979, Van Zandt & Berger, 2020) and a greater amplitude variation (Jiang & Pell, 2017; Van Zandt & Berger, 2020). Regarding pitch, findings have been inconsistent, with some studies finding that confident speech is uttered with higher pitch (see Van Zandt & Berger, 2020; Scherer et al., 1973) and others that confident speech is uttered with lower pitch (Apple et al., 1979; Guyer et al., 2019; Jiang & Pell, 2017). It is worthwhile to examine vocal characteristics and confidence in an eyewitness testimony setting, to determine to what extent these results generalize to a forensic setting.

This paper asks two research questions: First, do speakers use different vocal cues depending on their level of confidence? Second, do listeners perceive these potential vocal cues to confidence?

We have also added a motivation for the replication in the Study 1 discussion:

p. 18, lines 12-19:

Thus, the current results strengthen the idea that non-verbal aspects of speech carry some information regarding the accuracy of the recalled information (see also Goupil & Aucouturier, 2021). However, it should be noted that we did not find the expected effect of higher pitch for correct statements, as originally found in Gustafsson et al. (2023). This suggests that pitch might not be a reliable cue for accuracy—which aligns with the contrasting finding regarding the relationship between pitch and confidence (e.g., Jiang & Pell, 2017; Van Zandt & Berger, 2020) —and highlights the importance of using big samples during replication attempts.

We also discuss findings relative to previous results:

pp. 18-19, lines 32-41:

However, not all predictions were supported, as we found no evidence for a greater amplitude variability (cf. Jiang & Pell, 2017; Van Zandt & Berger, 2020) nor for pitch variability (cf. Goupil et al., 2021; Jiang & Pell., 2017). A possible explanation could be that our study analyzed natural, spontaneous speech from eyewitness testimony, whereas the previous studies utilized voice samples from actors who were instructed to use pseudo words (Goupil et al., 2021), a specific speaking style (Jiang & Pell, 2017; Van Zandt & Berger, 2020), and/or specific sentences (Jiang & Pell, 2017). That said, as discussed above we did replicate several other effects from studies that had used acted speech. Overall, we find that higher speech rate, shorter pauses and greater amplitude are the most robust vocal characteristics for confident speech.

14. Additionally, the authors state that because the patterns of vocal characteristics for accuracy and confidence are similar, this means there is an association between confidence and accuracy. I do not really follow the logic here. Just because vocal characteristics for accuracy and confidence are similar does not necessarily mean that confidence is related to accuracy. Study 2 was by far more compelling in that it addressed a novel research question and seemed better supported by the introduction.

We appreciate this comment pushing back on our logic and agree that similar vocal characteristics for confidence and accuracy does not necessarily imply a direct connection between the two. However, it may suggest that they share some underlying features, as the alternative interpretation – that is, that 16 variables coincidentally behave similarly – seems less likely. Moreover, given the existing research showing that confidence and accuracy are often related, we believe our findings could contribute to this understanding. We have now clarified this in the Study 1 discussion:

p. 18, lines 24-32:

The results also partially replicated previous findings regarding the relationship between confidence and vocal characteristics, with more confident speech having a higher speech rate, shorter pauses and greater amplitude. This replicates previous studies (Apple et al., 1979; Goupil et al., 2021; Guyer et al., 2019; Jiang & Pell, 2017; Kimble & Seidel, 1991; Scherer et al., 1979, Van Zandt & Berger, 2020). Further, it indirectly demonstrates the tight association between confidence and

accuracy (e.g., Gustafsson et al., 2022; Juslin et al., 1996; Robinson et al., 1997; Spearing & Wade, 2022; Wixted & Wells, 2017), given that the patterns of vocal characteristics for accuracy and confidence are nearly identical (see Figures 1 and 2).

We have also added this to the General discussion:

pp. 27-28, lines 230-252:

Interestingly, the vocal characteristics that indicate accuracy overlapped almost completely with findings on witnesses' own self-reported confidence (see Figures 1 and 2). That is, when witnesses expressed being 100% confident in their accuracy (as compared to less than 100% confident) they had relatively similar vocal-characteristic levels as correct vs. incorrect statements. There is a known confidence-accuracy relationship in eyewitness research, with witnesses being more confident in correct memories (e.g., Spearing & Wade, 2022; Wixted & Wells, 2017), and the overlap when considering vocal cues is considerable. Although we cannot derive the causal relationship between speech and confidence, the current results clearly demonstrate that there is some attribute in a memory that simultaneously makes people feel confident and makes them speak in a specific manner in terms of acoustical properties—a manner which appears to be “straight forward”, as evidenced by a more fluent (fewer pauses), fast (higher speech rate), and loud (greater mean amplitude and greater amplitude in the first-formant region) speech. One could postulate that feeling confident makes one speak more fluently, quickly, and loudly, which are some aspects of speech that listeners consider when assessing a speaker's confidence (e.g., Apple et al., 1979; Guyer et al., 2019; Jiang & Pell, 2017). However, it is reasonable to assume that there are attributes even prior to the initial confidence that influence the speech—specifically, the underlying memory strength. Memory strength is an abstract term to describe a quality of the memory, which mainly involves how well encoded and easily retrieved it is. The ease with which a memory is retrieved has been shown to be a cue to confidence (Kelley & Lindsay, 1993), and we suggest that an easily retrieved memory will be uttered more fluently, that is with fewer and shorter pauses, and as a consequence, a higher speech rate. Thus, the strength of the memory likely affects both the confidence and the utterance of that memory.

15. The discussion of the manuscript was also quite short and did not discuss the practical implications of this work despite situating the study as applied in the abstract and introduction. The authors discuss that even though their design was conservative and there was no contextual information, participants were still able to judge accuracy. However, this is the opposite of the applied context the researchers are studying. In real cases, those judging testimonial accuracy will have contextual information and pre-existing beliefs. How would these results change in those circumstances? Are the vocals cues the authors studying able to be identified by listeners in everyday conversations and, if not, what do the authors think about how these results could be applied in a legal context? Answering questions like these would have strengthened the discussion.

We have now rewritten and expanded the discussion. This new version includes a section on practical implications and real cases (see also the response to Q#4):

pp. 29-30, lines 293-314:

As mentioned, judgment accuracy among the participants was modest at best, residing only slightly above chance level overall. Although this does take away from the potential applicability of the findings (i.e., it might not be worth the time and investment to teach people about the vocal cues to accuracy if their performance will not increase by more than a few percentage points), it is important to highlight the context of the data. That is, participants listened to ten randomized testimony statements from ten different witnesses, without knowing anything about the context of the crime that had been witnessed, nor anything about the questions that had probed the respective statement. Moreover, the audio clips (i.e., testimony statements) were generally quite short, with many lasting less than a second. Nevertheless, despite these minimal conditions, there was still valid signal coming through and participants found vocal cues to accuracy. Given their non-verbal nature, these cues may also allow for greater generalization and universality, compared to verbal cues, such as verbal confidence statements. However, this remains an empirical question for future studies.

It is also difficult to say to what extent vocal cues influence judgments of accuracy in real cases, where individuals who hear testimony also have contextual information and pre-existing beliefs (e.g., Garrett, 2011). Indeed, people likely judge accuracy based on a plethora of factors, such as semantics, phrasing, body language, and possibly clothing and race as well. These are not necessarily valid cues, and here we show that the vocal cues at least to a degree are valid and can be detected. Future studies can examine how big a role the vocal cues play, and in turn compare how big of a role they should play. Our findings should be examined further and replicated before laying down guidelines on how to evaluate eyewitness testimony in level of vocal characteristics.

p. 31, lines 325-334:

Not all hypotheses were supported. In Study 1, we did not find higher pitch for correct statements, nor greater amplitude variation and greater pitch variability for highly confident statements. This suggests that these vocal characteristics are less stable predictors of accuracy and confidence respectively, and that the focus of research for practical purposes should focus more on the robust predictors discussed above. In Study 2, we did not find that the Swedish sample had a better ability to detect accuracy in the vocal samples than the other cultural samples. If this finding can be replicated, it would suggest universal interpretations of speech accuracy. An important caveat however is that the accuracy predictions were rather modest (see Table 4), suggesting that verbal content should receive greater emphasis relative to vocal characteristics.

Reviewer comments: *Eyewitness Testimony Accuracy Is Conveyed Through Vocal Cues with Both Cultural Universals and Differences*

We appreciate this opportunity to resubmit a revised version of our manuscript and the helpful feedback that you as reviewers have provided. Below the original feedback appears in black font and our response in blue font. Please note that italicized texts refer to texts in the manuscript.

Update: The editors requested that we add statistical significance testing for all our data, which we have highlighted in the responses below, as well as in the main manuscript, with red text.

Reviewer #1:

Overall, this is a well-intentioned attempt to further examine the vocal/acoustic factors that affect eyewitness' verbal recollections of a witnessed event. In addition, the comparison of different cultures is worthwhile and addresses the generality of judgments across cultures. For both studies, the statistical analyses are complicated and time consuming to unpack, but they seem appropriate.

1. One issue I have is that the absence of support for three predictions was not addressed or explained either in the Study 1 Discussion or General Discussion. Do the authors have any possible explanations for the failure to support those predictions?

We have now expanded the discussion in Study 1, which now also includes a discussion about failed predictions:

p. 20, lines 261-266:

However, it should be noted that we did not find the expected effect of higher pitch for correct statements, as originally found in Gustafsson et al. (2023). This suggests that pitch might not be a reliable cue for accuracy—which aligns with the contrasting finding regarding the relationship between pitch and confidence (e.g., Jiang & Pell, 2017; Van Zandt & Berger, 2020)—and highlights the importance of using big samples during replication attempts.

pp. 20-21, lines 279-288:

However, not all predictions were supported, as we found no evidence for a greater amplitude variability (cf. Jiang & Pell, 2017; Van Zandt & Berger, 2020) nor for pitch variability (cf. Goupil et al., 2021; Jiang & Pell, 2017). A possible explanation could be that our study analyzed natural, spontaneous speech from eyewitness testimony, whereas the previous studies utilized voice samples from actors who were instructed to use pseudo words (Goupil et al., 2021), a specific speaking style (Jiang & Pell, 2017; Van Zandt & Berger, 2020), and/or specific sentences (Jiang & Pell, 2017). That said, as discussed above we did replicate several other effects from studies that had used acted speech. Overall, we find that higher speech rate, shorter

pauses and greater amplitude are the most robust vocal characteristics for confident speech.

2. In Study 2, we have the same issue again, which is the absence of explanation for hypotheses that were not supported. For example, do the authors have an explanation for why there was no overall difference in accuracy judgments across the three cultural groups?

We have expanded the discussion in Study 2 as well, which now also includes a discussion about failed predictions:

pp. 28, lines 456-461:

Surprisingly, we did not find that the participants performance differed significantly between the three cultural groups. This could possibly be due to the difficulty of the stimuli material, as many of the uttered statements were short in length and sometimes contained only one or a few words or syllables. This makes it more difficult to extract information from the statement, such as prosody (as well as semantic meaning for the Swedish participants). Future studies should examine cultural differences in testimony speech with longer utterances.

3. I also have an issue with the interpretation of the interactions between country and other variables related to the confidence judgments (lines 361-368; lines 372-378). I am not sure that the DIFFERENCE between correct and incorrect statements is the appropriate metric instead of just relying on the overall main effect of country. And in terms of comparing the magnitude of Pearson-R numbers and claiming that the numbers reflect a difference across groups, to the tune of 4.5 times greater, 13.5 times greater, and 3.0 times greater, it is my understanding this kind of comparison is prohibited. There are many reasons different correlation values are obtained in experiments and they do mean anything in terms of magnitudes of difference across groups.

We appreciate this opportunity to clarify an area where the previous version of the manuscript was unclear. The interaction term of accuracy x country on confidence indicates the extent to which participants would perceive correct statements as being uttered with higher confidence compared to incorrect statements, which is Hypothesis 2a. Relying on the main effect for country would provide information as to whether one of the cultures generally perceived *all* statements to have a greater level of confidence relative to how the other cultures perceived it, which is a very interesting different question.

To address the other important point regarding coefficient comparisons, we **now perform Z-tests** to allow comparisons of differences between the groups:

pp. 27, lines 433-437:

Partially supporting H3b, the difference between the Swedish and American samples was significant ($Z = 8.81$, 95% CI [6.85, 10.77], $p < .001$), but the difference between the American and the Indian sample did not reach statistical significance ($Z = 1.66$, 95% CI [-0.30, 3.62], $p = 0.098$).

4. I found the General Discussion to be brief. I would like to see more discussion of the failures of supporting certain hypotheses and what the impact of what those failures are, if any. I would also like to see more discussion of the practical ramifications of their findings on judgments of observers who listen to the eyewitness statements. This appears to be the

critical main thrust of the article. And finally, I wonder if the authors can identify any vocal/structural differences across the three languages that could have affected the results. I know that some tonal languages like Mandarin would be very different than English in terms of how vocal aspects of utterances are processed. This was mentioned in the Introduction (lines 102-104) but not addressed later.

We now discuss failed predictions in the general discussion, including practical ramifications:

p. 33, lines 582-591:

Not all hypotheses were supported. In Study 1, we did not find higher pitch for correct statements, nor greater amplitude variation and greater pitch variability for highly confident statements. This suggests that these vocal characteristics are less stable predictors of accuracy and confidence respectively, and that the focus of research for practical purposes should focus more on the robust predictors discussed above. In Study 2, we did not find that the Swedish sample had a better ability to detect accuracy in the vocal samples than the other cultural samples. If this finding can be replicated, it would suggest universal interpretations of speech accuracy. An important caveat however is that the accuracy predictions were rather modest (see Table 4), suggesting that verbal content should receive greater emphasis relative to vocal characteristics.

Regarding vocal/structural differences, we did not find any clear examples of culture-specific correlations between perceived confidence and vocal cues and now explicitly mention this in our discussion (“Do people attune to the vocal cues to accuracy?”):

pp. 31, lines 525-541:

We expected listeners in the Swedish sample to perform most accurately because they would also have access to the verbal content (despite being instructed to focus only on the nonverbal aspects of the speech), which meant they could judge accuracy by weighing that information, such as the likelihood of remembering a specific type of detail (e.g., a knife, or the color of the offender’s shoes). Swedish speakers would also be familiar with language-specific nonverbal vocal patterns (i.e., speech prosody; e.g., Fodor, 1998; Wagner & Watson, 2010), and also cultural specificity relating more generally to vocal expression (e.g., Laukka & Elfenbein, 2021). However, results showed no major differences between the language comprehension groups, possibly indicating that they relied on similar vocal cues. Indeed, such an idea is strengthened when examining the correlation between listener-perceived confidence in the witnesses’ speech, and the vocal characteristics of that speech. All language-comprehension groups had similar patterns of correlation coefficients, without clear cultural effects. The highest correlations were observed for loudness (Int [M], $r = .30$), energy in the first-formant region (F1 [amp], $r = .25$), speech rate (VoicedSegPerSec, $r = .18$) and pause length (UnvoicedSeg [M], $r = -.22$). We note that the strongest correlations were present for the vocal characteristics that best indicated accuracy, suggesting that participants did tune in to valid cues.

SPECIFIC COMMENTS

5. line 43 This is a general statement that needs some clarification

We have now rephrased the sentence to be clearer:

p. 3, lines 41-43:

There are many known cases whereby eyewitnesses have been wrong in their recall or identification, which has led to innocent convictions (Innocence project, n.d.), making research on witness accuracy important for social justice.

6. line 126 What was this previous research, possibly include citations.

We have now added references to this in the introduction:

p. 4, lines 83-86:

Regarding pitch, findings have been inconsistent, with some studies finding that confident speech is uttered with higher pitch (see Van Zandt & Berger, 2020; Scherer et al., 1973) and others that confident speech is uttered with lower pitch (Apple et al., 1979; Guyer et al., 2019; Jiang & Pell, 2017).

7. Lines 313-317 I wonder if giving the American and Indian samples this instruction created a confound. Would participants normally focus their attention on these aspects of the utterances in their own languages?

We now clarify that there is no confound because all three groups received the same instructions to focus on the nonverbal aspects of the voice and not on the meaning of the words when they made their judgment.

p. 24, lines 363-367:

Furthermore, all participants were informed that they should focus on the “nonverbal aspects of the speech, such as the tone of voice” rather than on the verbal content when making their judgments. The American and Indian participants were additionally informed that the statements were spoken in Swedish and they were not expected to understand the content of the speech.

8. Line 337 Table 4 – The false alarm rates are very high, do the authors have any explanation for this outside just saying it's a liberal bias?

We have now added a bit about this in the general discussion:

p. 30-3§1, lines 515-522:

Participants also displayed a clear “truth bias,” as evident from the high false alarm rate (see Table 4). Research finds a similar effect in lie detection, that is, a tendency for people to be better at detecting lies than truth (see e.g., Vrij & Baxter, 1999). This liberal bias obtained here might result from an availability heuristic, where being exposed in daily life l mostly to correct statements makes individuals more inclined to believe another person’s recollection. This in turn suggests that the many real-life cases of mistaken eyewitnesses (see Innocenceproject, n.d.) might not have a large influence on beliefs about eyewitness accuracy.

Reviewer #2:

9. While I appreciate the authors' attempt to keep the manuscript concise and understand journal formatting guidelines, the brevity in the manuscript caused problems in each section.

We have now expanded each section of the manuscript to increase the narrative, depth and clarity of the text.

10. The introduction of the manuscript felt very jumbled. It was difficult to find a streamlined narrative that justified the current studies. Rather, it felt that topics jumped around frequently and thus, by the end of the introduction, the justification for the current studies felt weak and some of the past literature was discussed in an overly broad manner. For instance, in discussing eyewitness confidence, the authors cite the Wixted and Wells (2017) review paper. However, this paper focuses on eyewitness identification confidence and not eyewitness testimonial confidence. Moreover, the relationship between confidence and accuracy discussed in that paper is not absolute, but rather depends on the circumstances of the lineup. The authors make claims related to confidence like "nonverbal cues for confidence levels can vary across cultures and reduce cross-cultural accuracy" but do not provide support for this claim. The authors state that "speaker confidence is a potentially valid cue from speech research that is worth expanding to study in the eyewitness context". I would have liked to see them expand on this statement to really bring home the point about why this research is needed.

We have now rewritten the introduction with an attempt to build a stronger narrative, including more specific effects in the literature. For example, regarding confidence, we have now added both references to eyewitness identifications and testimony alike (as they both show same trend of effects), but added that correct memories are not always rated with higher confidence:

p. 4, lines 65-68:

Moreover, the witness generally rates themselves as more confident when their memories are correct (albeit not always), both eyewitness testimony and line-up identifications (e.g., Gustafsson et al., 2022; Juslin et al., 1996; Robinson et al., 1997; Spearing & Wade, 2022; Wixted & Wells, 2017).

We have also expanded the introduction to more strongly motivate research on the relationship between vocal characteristics of confident speech in eyewitness testimony:

pp. 4-5, lines 75-91:

Although there are few studies on vocal characteristics and eyewitness accuracy (Goupil & Aucouturier, 2021; Gustafsson et al., 2023), considerable evidence exists on the relationship between vocal characteristics and confidence more generally. Specifically, studies find that speech is judged (and produced) as more persuasive and confident when uttered with a higher speech rate (including shorter pauses; Apple et al., 1979; Goupil et al., 2021; Guyer et al., 2019; Kimble & Seidel, 1991; Scherer et al., 1979, Van Zandt & Berger, 2020), greater amplitude (Jiang & Pell, 2017; Kimble & Seidel, 1991; Scherer et al., 1979, Van Zandt & Berger, 2020) and a greater amplitude variation (Jiang & Pell, 2017; Van Zandt & Berger, 2020). Regarding pitch, findings have been inconsistent, with some studies finding that confident speech is uttered with higher pitch (see Van Zandt & Berger, 2020; Scherer et al., 1973) and others that confident speech is uttered with lower pitch (Apple et al., 1979; Guyer et

al., 2019; Jiang & Pell, 2017). It is worthwhile to examine vocal characteristics and confidence in an eyewitness testimony setting, to determine to what extent these results generalize to a forensic setting.

This paper asks two research questions: First, do speakers use different vocal cues depending on their level of confidence? Second, do listeners perceive these potential vocal cues to confidence?

11. In Study 1, the authors measure various aspects of vocal characteristics of eyewitness testimonial statements. However, the authors do not explain what these characteristics are or why they are chosen. For instance, what is the first formant region and why are the authors predicting this will be associated with correct statements. Speech researchers might be familiar with these vocal cues but memory researchers or criminal justice researchers likely will not.

Thank you for highlighting this. We understand that many of the acoustic variables are rather technical and difficult to unpack. Indeed, many of these vocal characteristics are difficult to explain in layman terms while retaining technical accuracy. To mitigate this, we categorize them into four overarching groups, that *can* generally be explained in more easily understood terms. These terms are *frequency, energy, spectral balance, and temporal characteristics*. We have clarified this categorization of the acoustic variables, and added descriptions of them:

p. 8, lines 171-178:

The audio files were analyzed along the 16 acoustic dimensions used in Gustafsson et al. (2023). Gustafsson et al. (2023) chose these dimensions by running a principal-component analysis on 88 acoustic dimensions, as well as by extrapolating findings from confidence speech research. These acoustic dimensions represent different aspects of frequency (e.g., fundamental frequency which is related to perceived pitch), energy (related to perceived loudness), spectral balance (related to perceived voice quality), and temporal characteristics (e.g., speech rate and pauses) of the voice signal. Table 1 shows a description of each acoustic dimension (i.e., vocal characteristic).

Next, why might F1 (amp) – that is, relative energy of the spectral envelope in the first formant region – be related to correct statements? The straight-forward explanation is that it's backed by data: our previous study that explored relations between accuracy and vocal characteristics, found such a relationship. This is now more clearly outlined in the manuscript:

p. 4, lines 68-74:

Recently, Gustafsson et al. (2023) demonstrated that there are also nonverbal vocal cues to accuracy; correct statements in eyewitness testimony were uttered with higher pitch, greater energy in the first formant region, higher speech rate and shorter pauses (see also Goupil & Aucouturier, 2021). However, more studies are needed to assess the reliability of these initial findings. Moreover, existing research has not yet examined whether people can detect that these are vocal cues to accuracy.

Moreover, one could tentatively reflect that this relationship might occur because correct memories should theoretically have stronger memory traces (meaning a clearer image of the memory), and strong memories should be easier to verbalize than weak memories. This stronger verbalization could manifest as a more loudly produced utterance, and F1 (amp) is

indeed a form of *Energy* acoustic variable. Moreover, well-defined, clear vowel sound are linked to the first-formant spectrum, which may explain why this effect is found in this specific spectrum. **The General discussion contains a brief discussion about the possible influence of memory strength on vocal characteristics (p. 30, lines 502-509).**

12. I also found it concerning that the authors placed the effect sizes in the supplemental materials as I think this is a needed statistic in the main manuscript (in Study 2, effect sizes are also missing for several analyses).

We have now added effect sizes to the figures, and also added effect sizes **(with confidence intervals) to all analyses reported in the manuscript, including the one-sample *t*-tests in Study 2:**

pp. 24-25, lines 379-385:

*We tested this with one-sample *t*-tests against 0 ($d' = 0$ indicates chance performance). In line with predictions (H1a), all groups performed above chance level; Swedish sample: $t(60) = 3.98, p < .001$, Cohen's $d = 0.51$, 95% CI [0.24, 0.78]; American sample: $t(155) = 5.94, p < .001$, Cohen's $d = 0.48$, 95% CI [0.31, 0.64]; Indian sample: $t(59) = 5.09, p < .001$, Cohen's $d = 0.66$, 95% CI [0.37, 0.94]; with an overall score across groups above chance level: $t(276) = 8.47, p < .001$, Cohen's $d = 0.51$, 95% CI [0.38, 0.63] (see Table 4).*

13. The discussion section for study 1 is also very short and does not effectively communicate what the contribution of this study is beyond replicating past work. If the only goal of this study was to replicate past work, more detail is needed about why this replication was necessary and how this replication adds to existing knowledge.

The motivation for Study 1 has now more clearly been added to the introduction:

pp. 4-5, lines 68-91:

Recently, Gustafsson et al. (2023) demonstrated that there are also nonverbal vocal cues to accuracy; correct statements in eyewitness testimony were uttered with higher pitch, greater energy in the first formant region, higher speech rate and shorter pauses (see also Goupil & Aucouturier, 2021). However, more studies are needed to assess the reliability of these initial findings. Moreover, existing research has not yet examined whether people can detect that these are vocal cues to accuracy.

Although there are few studies on vocal characteristics and eyewitness accuracy (Goupil & Aucouturier, 2021; Gustafsson et al., 2023), considerable evidence exists on the relationship between vocal characteristics and confidence more generally. Specifically, studies find that speech is judged (and produced) as more persuasive and confident when uttered with a higher speech rate (including shorter pauses; Apple et al., 1979; Goupil et al., 2021; Guyer et al., 2019; Kimble & Seidel, 1991; Scherer et al., 1979, Van Zandt & Berger, 2020), greater amplitude (Jiang & Pell, 2017; Kimble & Seidel, 1991; Scherer et al., 1979, Van Zandt & Berger, 2020) and a greater amplitude variation (Jiang & Pell, 2017; Van Zandt & Berger, 2020). Regarding pitch, findings have been inconsistent, with some studies finding that confident speech is uttered with higher pitch (see Van Zandt & Berger, 2020; Scherer et al., 1973) and others that confident speech is uttered with lower pitch (Apple et al., 1979; Guyer et al., 2019; Jiang & Pell, 2017). It is worthwhile to examine vocal

characteristics and confidence in an eyewitness testimony setting, to determine to what extent these results generalize to a forensic setting.

This paper asks two research questions: First, do speakers use different vocal cues depending on their level of confidence? Second, do listeners perceive these potential vocal cues to confidence?

We have also added a motivation for the replication in the Study 1 discussion:

p. 20, lines 259-266:

Thus, the current results strengthen the idea that non-verbal aspects of speech carry some information regarding the accuracy of the recalled information (see also Goupil & Aucouturier, 2021). However, it should be noted that we did not find the expected effect of higher pitch for correct statements, as originally found in Gustafsson et al. (2023). This suggests that pitch might not be a reliable cue for accuracy—which aligns with the contrasting finding regarding the relationship between pitch and confidence (e.g., Jiang & Pell, 2017; Van Zandt & Berger, 2020)—and highlights the importance of using big samples during replication attempts.

We also discuss findings relative to previous results:

pp. 20-21, lines 279-288:

However, not all predictions were supported, as we found no evidence for a greater amplitude variability (cf. Jiang & Pell, 2017; Van Zandt & Berger, 2020) nor for pitch variability (cf. Goupil et al., 2021; Jiang & Pell., 2017). A possible explanation could be that our study analyzed natural, spontaneous speech from eyewitness testimony, whereas the previous studies utilized voice samples from actors who were instructed to use pseudo words (Goupil et al., 2021), a specific speaking style (Jiang & Pell, 2017; Van Zandt & Berger, 2020), and/or specific sentences (Jiang & Pell, 2017). That said, as discussed above we did replicate several other effects from studies that had used acted speech. Overall, we find that higher speech rate, shorter pauses and greater amplitude are the most robust vocal characteristics for confident speech.

14. Additionally, the authors state that because the patterns of vocal characteristics for accuracy and confidence are similar, this means there is an association between confidence and accuracy. I do not really follow the logic here. Just because vocal characteristics for accuracy and confidence are similar does not necessarily mean that confidence is related to accuracy. Study 2 was by far more compelling in that it addressed a novel research question and seemed better supported by the introduction.

We appreciate this comment pushing back on our logic and agree that similar vocal characteristics for confidence and accuracy does not necessarily imply a direct connection between the two. However, it may suggest that they share some underlying features, as the alternative interpretation – that is, that 16 variables coincidentally behave similarly – seems less likely. Moreover, given the existing research showing that confidence and accuracy are often related, we believe our findings could contribute to this understanding. We have now clarified this in the Study 1 discussion:

p. 20, lines 271-279:

The results also partially replicated previous findings regarding the relationship between confidence and vocal characteristics, with more confident speech having a higher speech rate, shorter pauses and greater amplitude. This replicates previous studies (Apple et al., 1979; Goupil et al., 2021; Guyer et al., 2019; Jiang & Pell, 2017; Kimble & Seidel, 1991; Scherer et al., 1979, Van Zandt & Berger, 2020). Further, it indirectly demonstrates the tight association between confidence and accuracy (e.g., Gustafsson et al., 2022; Juslin et al., 1996; Robinson et al., 1997; Spearing & Wade, 2022; Wixted & Wells, 2017), given that the patterns of vocal characteristics for accuracy and confidence are nearly identical (see Figures 1 and 2).

We have also added this to the General discussion:

pp. 29-30, lines 487-509:

Interestingly, the vocal characteristics that indicate accuracy overlapped almost completely with findings on witnesses' own self-reported confidence (see Figures 1 and 2). That is, when witnesses expressed being 100% confident in their accuracy (as compared to less than 100% confident) they had relatively similar vocal-characteristic levels as correct vs. incorrect statements. There is a known confidence-accuracy relationship in eyewitness research, with witnesses being more confident in correct memories (e.g., Spearing & Wade, 2022; Wixted & Wells, 2017), and the overlap when considering vocal cues is considerable. Although we cannot derive the causal relationship between speech and confidence, the current results clearly demonstrate that there is some attribute in a memory that simultaneously makes people feel confident and makes them speak in a specific manner in terms of acoustical properties—a manner which appears to be “straight forward”, as evidenced by a more fluent (fewer pauses), fast (higher speech rate), and loud (greater mean amplitude and greater amplitude in the first-formant region) speech. One could postulate that feeling confident makes one speak more fluently, quickly, and loudly, which are some aspects of speech that listeners consider when assessing a speaker's confidence (e.g., Apple et al., 1979; Guyer et al., 2019; Jiang & Pell, 2017). However, it is reasonable to assume that there are attributes even prior to the initial confidence that influence the speech—specifically, the underlying memory strength. Memory strength is an abstract term to describe a quality of the memory, which mainly involves how well encoded and easily retrieved it is. The ease with which a memory is retrieved has been shown to be a cue to confidence (Kelley & Lindsay, 1993), and we suggest that an easily retrieved memory will be uttered more fluently, that is with fewer and shorter pauses, and as a consequence, a higher speech rate. Thus, the strength of the memory likely affects both the confidence and the utterance of that memory.

15. The discussion of the manuscript was also quite short and did not discuss the practical implications of this work despite situating the study as applied in the abstract and introduction. The authors discuss that even though their design was conservative and there was no contextual information, participants were still able to judge accuracy. However, this is the opposite of the applied context the researchers are studying. In real cases, those judging testimonial accuracy will have contextual information and pre-existing beliefs. How would these results change in those circumstances? Are the vocals cues the authors studying able to be identified by listeners in everyday conversations and, if not, what do the authors think

about how these results could be applied in a legal context? Answering questions like these would have strengthened the discussion.

We have now rewritten and expanded the discussion. This new version includes a section on practical implications and real cases (see also the response to Q#4):

p. 32, lines 550-571:

As mentioned, judgment accuracy among the participants was modest at best, residing only slightly above chance level overall. Although this does take away from the potential applicability of the findings (i.e., it might not be worth the time and investment to teach people about the vocal cues to accuracy if their performance will not increase by more than a few percentage points), it is important to highlight the context of the data. That is, participants listened to ten randomized testimony statements from ten different witnesses, without knowing anything about the context of the crime that had been witnessed, nor anything about the questions that had probed the respective statement. Moreover, the audio clips (i.e., testimony statements) were generally quite short, with many lasting less than a second. Nevertheless, despite these minimal conditions, there was still valid signal coming through and participants found vocal cues to accuracy. Given their non-verbal nature, these cues may also allow for greater generalization and universality, compared to verbal cues, such as verbal confidence statements. However, this remains an empirical question for future studies.

It is also difficult to say to what extent vocal cues influence judgments of accuracy in real cases, where individuals who hear testimony also have contextual information and pre-existing beliefs (e.g., Garrett, 2011). Indeed, people likely judge accuracy based on a plethora of factors, such as semantics, phrasing, body language, and possibly clothing and race as well. These are not necessarily valid cues, and here we show that the vocal cues at least to a degree are valid and can be detected. Future studies can examine how big a role the vocal cues play, and in turn compare how big of a role they should play. Our findings should be examined further and replicated before laying down guidelines on how to evaluate eyewitness testimony in level of vocal characteristics.

p. 33, lines 582-591:

Not all hypotheses were supported. In Study 1, we did not find higher pitch for correct statements, nor greater amplitude variation and greater pitch variability for highly confident statements. This suggests that these vocal characteristics are less stable predictors of accuracy and confidence respectively, and that the focus of research for practical purposes should focus more on the robust predictors discussed above. In Study 2, we did not find that the Swedish sample had a better ability to detect accuracy in the vocal samples than the other cultural samples. If this finding can be replicated, it would suggest universal interpretations of speech accuracy. An important caveat however is that the accuracy predictions were rather modest (see Table 4), suggesting that verbal content should receive greater emphasis relative to vocal characteristics.